# Estimates of Ozone Return Dates from Chemistry-Climate Model Initiative Simulations

Sandip S. Dhomse (1), Douglas Kinnison (2), Martyn P. Chipperfield (1,3), Ross J. Salawitch (4,5,6), Irene Cionni (7), Michaela I. Hegglin (8), N. Luke Abraham (9,10), Hideharu Akiyoshi (11), Alex T. Archibald (9,10), Ewa M. Bednarz (9), Slimane Bekki (12), Peter Braesicke (13), Neal Butchart (14), Martin Dameris (15), Makoto Deushi (16), Stacey Frith (17, 18), Steven C. Hardiman (14), Birgit Hassler (15), Larry W. Horowitz (19), Rong-Ming Hu (12), Patrick Jöckel (15), Beatrice Josse (20), Oliver Kirner (21), Stefanie Kremser (22), Ulrike Langematz (23), Jared Lewis (22), Marion Marchand (12), Meiyun Lin (19,24), Eva Mancini (25), Virginie Marécal (20), Martine Michou (20), Olaf Morgenstern (26), Fiona M. O'Connor (14), Luke Oman (18), Giovanni Pitari (27), David A. Plummer (28), John A. Pyle (9,10), Laura E. Revell (22,29), Eugene Rozanov (29,30), Robyn Schofield (31,32), Andrea Stenke (29), Kane Stone (31,32)*, Kengo Sudo (33,34), Simone Tilmes (2), Daniele Visioni (25), Yousuke Yamashita (11,34), Guang Zeng (26)

1) School of Earth and Environment, University of Leeds, Leeds, LS2 9JT, U.K.
2) National Center for Atmospheric Research (NCAR), Boulder, Colorado, USA.
3) National Centre for Earth Observation, University of Leeds, LS2 9JT, UK.
4) Department of Chemistry and Biochemistry, University of Maryland, College Park, Maryland, USA.
5) Department of Atmospheric and Oceanic Science, University of Maryland, College Park, Maryland, USA.
6) Earth System Science Interdisciplinary Center, University of Maryland, College Park, Maryland, USA
7) Agenzia Nazionale per le Nuove Tecnologie, l'energia e lo Sviluppo Economica Sostenible (ENEA), Bologna, Italy.
8) Department of Meteorology, University of Reading, Reading, U.K.
9) Department of Chemistry, University of Cambridge, Lensfield Road, Cambridge, CB2 1EW, U.K.
10) National Centre for Atmospheric Science, U.K.
11) National Institute for Environmental Studies (NIES), Tsukuba, 305-8506 Japan.
12) IPSL/CNRS, 75252 Paris, France.
13) IMK-ASF, KIT, Karlsruhe, Germany.
14) Met Office Hadley Centre, Exeter, U.K.
15) Deutsches Zentrum fur Luft- und Raumfahrt (DLR), Institut fur Physik der Atmosphare, Oberpfaffenhofen, Germany.
16) Meteorological Research Institute (MRI), Tsukuba, Japan.
17) Science Systems and Applications, Inc., Lanham, MD, USA.
18) NASA/GSFC, USA.
19) NOAA Geophysical Fluid Dynamics Laboratory, Princeton, NJ 08540, USA.
20) Meteo-France, Toulouse, France.
21) Steinbuch Centre for Computing (SCC), Karlsruhe Institute of Technology (KIT), Karlsruhe, Germany.
22) Bodeker Scientific, Alexandra, New Zealand.
23) Institut für Meteorologie, Freie Universitat Berlin, Berlin, Germany.
24) Atmospheric and Oceanic Sciences, Princeton University, Princeton, NJ 08540, USA.
25) Dept. of Physical and Chemical Sciences & Center of Excellence CETEMPS, Universita del l'Aquila, Italy.
26) National Institute of Water and Atmospheric Research (NIWA), Wellington, New Zealand.
27) Department of Physical and Chemical Sciences, Universitat dell'Aquila, Italy.
28) Climate Research Division, Environment and Climate Change Canada, Montreal, Canada.
29) ETH Zurich, Institute for Atmospheric and Climate Science, Zurich, Switzerland.
30) Physikalisch-Meteorologisches Observatorium Davos World Radiation Centre, Davos Dorf, Switzerland.
31) School of Earth Sciences, University of Melbourne, Melbourne, Australia.
32) ARC Centre of Excellence for Climate System Science, Sydney, Australia.
33) Graduate School of Environmental Studies, Nagoya University, Nagoya, Japan.
34) Japan Agency for Marine-Earth Science and Technology (JAMSTEC), Yokohama, 236-0001 Japan.
* Now at Massachusetts Institute of Technology (MIT), Boston, Massachusetts, USA.

Revised for Atmos. Chem. Phys. May 24, 2018

*Correspondence to*: S.S.Dhomse@leeds.ac.uk, dkin@ucar.edu and M.Chipperfield@leeds.ac.uk

**Abstract.** We analyse simulations performed for the Chemistry-Climate Model Initiative (CCMI) to estimate the return dates of the stratospheric ozone layer from depletion caused by anthropogenic stratospheric chlorine and bromine. We consider a total of 155 simulations from 20 models, including a range of sensitivity studies which examine the impact of climate change on ozone recovery. For the control simulations (unconstrained by nudging towards analysed meteorology) there is a large spread ($\pm$20 DU in the global average) in the predictions of the absolute ozone column. Therefore, the model results need to be adjusted for biases against historical data. Also, the interannual variability in the model results need to be smoothed in order to provide a reasonably narrow estimate of the range of ozone return dates. Consistent with previous studies, but here for a Representative Concentration Pathway (RCP) of 6.0, these new CCMI simulations project that global total column ozone will return to 1980 values in 2049 (with a 1-$\sigma$ uncertainty of 2043-2055). At Southern Hemisphere mid-latitudes column ozone is projected to return to 1980 values in 2045 (2039-2050), and at Northern Hemisphere mid-latitudes in 2032 (2020-2044). In the polar regions, the return dates are 2060 (2055-2066) in the Antarctic in October and 2034 (2025-2043) in the Arctic in March. The earlier return dates in the NH reflect the larger sensitivity to dynamical changes. Our estimates of return dates are later than those presented in the 2014 Ozone Assessment by approximately 5-17 years, depending on the region, with the previous best estimates often falling outside of our uncertainty range. In the tropics only around half the models predict a return of ozone to 1980 values, at around 2040, while the other half do not reach the 1980 value. All models show a negative trend in tropical total column ozone towards the end of the 21$^{st}$ century. The CCMI models generally agree in their simulation of the time evolution of stratospheric chlorine and bromine, which are the main drivers of ozone loss and recovery. However, there are a few outliers which show that the multi-model mean results for ozone recovery are not as tightly constrained as possible. Throughout the stratosphere the spread of ozone return dates to 1980 values between models tends to correlate with the spread of the return of inorganic chlorine to 1980 values. In the upper stratosphere, greenhouse gas-induced cooling speeds up the return by about 10-20 years. In the lower stratosphere, and for the column, there is a more direct link in the timing of the return dates of ozone and chlorine, especially for the large Antarctic depletion. Comparisons of total column ozone between the models is affected by different predictions of the evolution of tropospheric ozone within the same scenario, presumably due to differing treatment of tropospheric chemistry. Therefore, for many scenarios, clear conclusions can only be drawn for stratospheric ozone columns rather than the total column. As noted by previous studies, the timing of ozone recovery is affected by the evolution of $N_2O$ and $CH_4$. However, quantifying the effect in the simulations analysed here is limited by the few realisations available for these experiments compared to internal model variability. The large increase in $N_2O$ given in RCP 6.0 extends the ozone return globally by ~15 years relative to $N_2O$ fixed at 1960 abundances, mainly because it allows tropical column ozone to be depleted. The effect in extratropical latitudes is much smaller. The large increase in $CH_4$ given in the RCP 8.5 scenario compared to RCP 6.0 also lengthens ozone return by ~15 years, again mainly through its impact in the tropics. Overall, our estimates of ozone return dates are uncertain due to both uncertainties in future scenarios, in particular of greenhouse gases, and uncertainties in models. The scenario uncertainty is small in the short term but increases with time, and becomes large by the end of the century. There are still some model-model differences related to well-known processes which affect ozone recovery. Efforts need to continue to ensure that models used for assessment purposes accurately represent stratospheric chemistry and the prescribed scenarios of ozone-depleting substances, and only those models are used to calculate return dates. For future assessments of single forcing or combined effects of $CO_2$, $CH_4$, and $N_2O$ on the stratospheric column ozone return dates, this work suggests that is more important to have multi-member (at least 3) ensembles for each scenario from every established participating model, rather than a large number of individual models.

## 1) Introduction

It is well established that stratospheric ozone depletion over the past few decades has been mostly driven by the increase in stratospheric chlorine and bromine originating from the use of halogenated ozone-depleting substances (ODSs). The production of these ODSs has now been largely controlled by the 1987 Montreal Protocol on Substances that Deplete the Ozone Layer, and its subsequent amendments and adjustments. For this reason, the protocol has been recognised as one of the most successful international environmental treaties ever enacted. Following its implementation, the atmospheric burdens of many ODSs have peaked and are now declining at a rate dependent on their respective atmospheric lifetimes, which are typically many decades. This has led to the overall levels of tropospheric chlorine and bromine peaking in 1993 and 1998, respectively (WMO, 2014). These levels are now slowly decreasing and are expected to return to their 1980 values, an arbitrary reference date before the discovery of the Antarctic ozone hole, around the middle of this century. Accordingly, stratospheric ozone is expected to recover from the effects of halogen-induced decreases on a similar timescale, which is already detectable in the Antarctic and upper stratosphere (e.g. see *Solomon et al.,* 2016; *Chipperfield et al*., 2017; *Strahan and Douglass*, 2018 and references therein). However, other atmospheric changes, notably climate change through increasing levels of greenhouse gases (GHGs) are also expected to modify the rate of return of ozone and its subsequent evolution (e.g., *Shepherd and Jonsson*, 2008; *Eyring et al*., 2010a). The effect of this climate impact is likely to be different in various latitudinal and altitudinal regions of the stratosphere and, for the polar regions, depends on the dynamical evolution of the polar vortices.

The prediction of ozone recovery and return therefore requires the use of three-dimensional (3D) coupled chemistry-climate models (CCMs) which contain details of the important stratospheric chemical processes, as well as a realistic, interactive representation of stratospheric temperature and dynamics (*Morgenstern et al*., 2010, 2017 and references therein). In these models, the simulated composition of the atmosphere is fully interactive, wherein the radiatively active gases, e.g., carbon dioxide ($CO_2$), water vapour ($H_2O$), nitrous oxide ($N_2O$), methane ($CH_4$), chlorofluorocarbons (CFCs), and ozone ($O_3$) affect model heating and cooling rates and therefore dynamics. The representation of stratospheric chemistry includes species and chemical reactions contained in the odd oxygen ($O_x$), nitrogen ($NO_x$), hydrogen ($HO_x$), chlorine ($ClO_x$), and bromine ($BrO_x$) chemical families as recommended, for example, by the NASA Chemical Kinetics and Photochemical Data for Use in Atmospheric Studies evaluation reports (e.g., *Sander et al.,* 2011). These chemical recommendations also include heterogeneous processes on sulfate aerosols and polar stratospheric clouds (PSCs). In addition, many of the CCMs include a detailed representation of tropospheric chemistry.

A process-oriented evaluation of CCMs was performed as part of the Stratospheric Processes And their Role in Climate (SPARC) Chemistry Climate Model Validation Activity (CCMVal). Quantitative performance metrics (*Eyring et al.,* 2008; *Waugh and Eyring* 2008) were used in an extensive effort to evaluate the ability of CCMs to represent key processes related to dynamics, transport, chemistry, and climate. These CCMs have been used in support of several international assessments of ozone depletion and recovery (e.g., *Austin et al*., 2003, 2010; WMO/UNEP 2003, 2007, 2011, 2014; *Eyring, et al*., 2007, 2010a, b, 2013b). The models used in this current paper have benefitted from the testing and development that came out of these studies and many are direct updates of the models tested in CCMVal.

Radiative, dynamical and chemical processes affect ozone recovery and return date. The radiative effects of changing GHG concentrations and ozone-induced temperature changes can affect the wave driving of the atmosphere and subsequently can accelerate the Brewer-Dobson (*Brewer,* 1949) (BD) circulation (*Rind et al.*, 1990, 2001; *Butchart and Scaife*, 2001; *Sigmond et al.*, 2004,
*Eichelberger and Hartmann,* 2005; *Butchart et al.*, 2006, 2010; *Olsen et al.,* 2007; *Garcia and Randel,* 2008; *McLandress et al.,* 2010, *Polvani et al.,* 2017, 2018). These studies have shown that the acceleration of the BD circulation is a robust result across the majority of CCMs. This acceleration can affect ozone recovery by causing faster removal of ODSs and hence earlier ozone recovery, although decreases in tropical stratospheric ozone may result in column ozone remaining
permanently below the 1980 value at these latitudes. *Butchart and Scaife* (2001) estimated that the recovery of the global ozone layer could be brought forward by 8-10 years. This shortening of the ozone recovery was also found by *Morgenstern et al.* (2018) for the models represented in this study, although it is important to note that the use of surface mixing ratios in studies largely removes the feedback between circulation changes and ODS return dates. In addition, this
strengthening of the BD circulation leads to a decrease in the mean age of stratospheric air by about 0.05 years per decade (*Butchart et al.,* 2010).

Increases in GHGs lead to cooling of the upper stratosphere (e.g., *Fels et al.*, 1980; *Rind et al.*, 1990) which slows down temperature-dependent odd-oxygen loss processes and increases upper
stratospheric ozone (e.g., *Haigh and Pyle*, 1982; *Brasseur and Hitchman*, 1988; *Pitari et al.,* 1992; *Rosenfeld et al.,* 2002; *Waugh et al.*, 2009; *Oman et al.*, 2010; *Bekki et al.*, 2013; *Marsh et al.,* 2016). Evolution of $N_2O$ and $CH_4$ can also impact ozone recovery by both chemistry and climate processes (*Randeniya et al.,* 1997; *Chipperfield and Feng*, 2003; *Ravishankara et al.,* 2009; *McLandress et al.*, 2010; *Revell et al.*, 2012; *Morgenstern et al.,* 2014; *Revell et al.*, 2015; *Kirner*
*et al.*, 2015).

A number of studies have used detailed coupled CCMs to predict the future evolution of stratospheric column ozone. For example, using the GFDL model, *Austin and Wilson* (2006) showed that the October monthly mean Antarctic ozone return to 1980 abundances mainly
depends on halogen loading and will not occur until ~2065. They also showed that Arctic ozone returns about 25-35 years earlier than Antarctic ozone does, while *Li et al*. (2009) showed that in the tropics ozone may never return to 1980 values due to changes in transport. However, due to various biases and weaknesses amongst various CCMs, our best estimates of ozone return dates come from multi-model assessments such as CCMI.
The aim of this paper is to provide updated estimates of the return dates of stratospheric ozone in different latitude regions. The year at which column ozone is expected to return to a particular, historic level is a key metric for quantifying the recovery of the ozone layer (e.g., Table 2-5, WMO 2014). Note that we estimate return dates based on mean atmospheric behaviour, independent of
40 the extremes caused by dynamically driven variability, for example. Our estimates of return dates are based on analysis of the new, extensive range of CCMI simulations which have been produced using updated and improved models compared to previous studies such as CCMVal-2. All models compute the impact on ozone of future stratospheric cooling, the intensification of the Brewer-Dobson circulation, as well as numerous other dynamical and chemical factors. Three types of
45 uncertainties are considered: internal variability, model uncertainty, and scenario uncertainty. Internal variability refers to short-term variations in computed ozone, which are not dependent on long-term change driven by ODSs and GHGs. Model uncertainty represents the fact that various CCMs can provide different return dates, for the same prescribed time series of ODSs and GHGs, due to alternate representations of processes such as the impact of rising GHGs on stratospheric
circulation. Finally, scenario uncertainty means that the return date found by a CCM will be sensitive to the particular, prescribed time series of ODSs and GHGs used as model input.

The layout of the paper is as follows. **Section 2** briefly summarises the models, scenarios, and simulations used in this study. **Section 3** describes how the results of the CCM simulations are processed in order to smooth over interannual variability and adjust some model ozone values for
biases compared to observations. **Section 4** presents our main results on the return and recovery of stratospheric ozone and how it varies between different simulations and between models. Finally, **Section 5** presents our conclusions.

## 2) Models, Scenarios, and Simulations

The model simulations used in this work are taken from the Chemistry–Climate Model Initiative (CCMI), which is a joint activity from the International Global Atmospheric Chemistry (IGAC) project and SPARC. One of the main goals of CCMI is to provide support for the quadrennial Scientific Assessment of Ozone Depletion, produced by the World Meteorological Organisation (WMO) Global Ozone Research and Monitoring Project and United Nations Environment
Programme (UNEP). This is done through the provision of scenarios and input forcings that the models can use for standard experiments. This paper provides analysis to support the results that will be presented in the forthcoming 2018 assessment.

Descriptions of the CCMI experiments used in this work are given in **Table 1.** These scenarios
are discussed in detail in *Eyring et al.* (2013a) and *Morgenstern et al.* (2017). **Table 2** summarises the simulations performed by each model considered here. Briefly, there are three reference simulations designed to understand both past and future ozone evolution. The first (labelled REF-C1) is a hindcast simulation of the recent past [1960-2010] that is closely tied to the following observed time-dependent forcings: 1) greenhouse gases (GHGs) and ozone-depleting substances
(ODSs); 2) 11-year solar variability; 3) sulfate aerosol surface area density (including background and volcanically active periods); 4) sea surface temperatures (SSTs) and sea ice concentrations (SICs); 5) additional organic bromine from very short-lived substances (VSLS); and 6) tropospheric ozone and aerosol precursor emissions. The meteorology in REF-C1 is free-running. The second reference simulation (labelled REF-C1SD) has the same observed forcings as REF-
C1, with the additional constraint that model temperature and dynamics are nudged to analysed meteorology, i.e., specified dynamics (SD). The third reference scenario (labelled REF-C2) includes both a hindcast and forecast period [1960-2100]. It should be noted that for REF-C2, several of the models used in this study have an interactive ocean/sea ice modules (*Morgenstern et al.*, 2018). For this scenario, the hindcast forcings are similar to REF-C1 with the main exception
that the SSTs/SICs are based on model results (*Morgenstern et al.*, 2018). The forecast component of the REF-C2 scenario uses GHGs (i.e., $CO_2$, $CH_4$, and $N_2O$) that follow the Intergovernmental Panel on Climate Change (IPCC) Coupled Model Intercomparison Project Phase 5 (CMIP5) Representative Concentration Pathways 6.0 (RCP 6.0) scenario (*Meinshausen et al.*, 2011; *Masui et al.*, 2011).

**Table 2** shows the number of realisations that were available for each simulation from every model. For REF-C1 a total of 19 models performed 38 realisations, including 8 models which performed multi-member ensembles. For simulations REF-C1SD and REF-C2 the numbers are 13 models (13 realisations) and 19 models (33 realisations), respectively. In addition to these
reference simulations, eight additional sensitivity scenarios based on the REF-C2 simulation were defined by CCMI. SEN-C2-RCP26 (5 models, 5 realisations), SEN-C2-RCP45 (7, 9) and SEN-C2-RCP85 (8, 10) follow RCP 2.6, 4.5 and 8.5, respectively. These scenarios diverge from the REF-C2 definition in year 2000. SEN-C2-fODS (8, 12) and SEN-C2-fGHG (8, 13) are identical to REF-C2, except that concentrations of ODSs and GHGs, respectively, are fixed at 1960 levels.

Finally, there are three scenarios that examine the sensitivity of ozone return to $N_2O$ and $CH_4$. Scenario SEN-C2-fN2O [1960-2100] is the same as REF-C2 but with surface $N_2O$ mixing ratios fixed to 1960 values. This results in the $N_2O$ surface abundance being ~40% higher in 2100 for the REF-C2 scenario versus the SEN-C2-fN2O scenario. Scenario SEN-C2-fCH4 [1960-2100] is the same as REF-C2 but with surface $CH_4$ abundance fixed to 1960 values; the late 21st century REF-C2 surface abundance of $CH_4$ is 30-50% higher than in SEN-C2-fCH4. Scenario SEN-C2-CH4RCP85 [2000-2100] replaces the REF-C2 RCP 6.0 $CH_4$ surface abundance with that from RCP8.5. Since the RCP8.5 surface $CH_4$ is considerably larger than in REF-C2, there is 110-125% more $CH_4$ in SEN-C2-CH4RCP85 relative to REF-C2 in the late 21st century. There are 8 (8), 8 (8), 6 (6) CCMs (realisations) included for SEN-C2-fN2O, SEN-C2-fCH4 and SEN-C2-CH4RCP85, respectively. For scenarios SEN-C2-RCP26 and SEN-C2-CH4RCP85, in particular, the numbers of realisations available are limited.

### 3) Methodology

We present an analysis based on seven latitude bands: Southern Hemisphere (SH) polar (90ºS-60ºS), SH mid-latitudes (60ºS-35ºS), tropics (20ºS-20ºN), Northern Hemisphere (NH) mid-latitudes (35ºN-60ºN), NH polar (60ºN-90ºN), near-global (60ºS-60ºN) and global (90ºS-90ºN). For the SH and NH polar regions we only consider the months of October and March, respectively, which typically correspond to the end of the winter/spring ozone loss periods. For the other latitude bands, we use annual means from each model simulation. We use 60ºS-60ºN for the near-global means, and 90ºS-90ºN for the global means, to allow for various latitudinal definitions of "global" ozone analyses in WMO ozone assessments (e.g. WMO 2011; WMO, 2014). In their analysis of CCMVal-2 simulations, *Eyring et al*. (2010b) used time-series additive model (TSAM) (*Scinocca et al*., 2009) but here we show absolute ozone time series to identify models to be excluded as outliers. As shown in **Table 2**, there are variations in the number of simulations for each reference/sensitivity experiment. Therefore, we decided to calculate the model averages (means and median) for each scenario using the following procedure.

- First, we calculate the zonal mean October, March or annual mean time series for each realisation. If more than one realisation is available for a particular model then we calculate an ensemble mean of monthly/annual mean time series. If only one realisation is available then we apply a 3-point (i.e. 3-year) boxcar smoothing to remove the short-term variations and somewhat mimic the effect of averaging an ensemble.

- These single time series for individual models are used to calculate the multi-model mean (MMM) and corresponding standard deviations for each year. We also calculate the '1-σ multi-model mean' (MMM1S) where we exclude models lying outside 1σ of the MMM for each year. Finally, all of the single time series are used to calculate the median model (MedM) along with the 10th and 90th percentiles.

- To calculate smoothed and adjusted time series with respect to a reference year (e.g. 1960, 1980), we apply a 10-point boxcar smoothing to the individual model time series as well as the MMM, MMM1S and MedM time series.

- In order to make a robust estimate of ozone return dates we need to account for the different biases between the model simulations and observations. We calculate the mean biases between observational data and the REF-C2 MMM, MMM1S and MedM timeseries for the 1980-1984 time period. We choose the REF-C2 simulation as that is the reference that is used to estimate ozone return dates. An adjusted timeseries for each individual model is

then calculated by subtracting the respective observational bias. This procedure also results in the multi-model mean agreeing with the observations in the 1980-1984 period.

- Finally, using the MMM1S timeseries from REF-C2 as a reference line, the MMM1S timeseries from REF-C1, REF-C1SD, SEN-C2-fODS, SEN-C2-fGHG, SEN-C2-fCH4 and SEN-C2-fN2O are adjusted for the year 1980. The MMM1S timeseries from simulations starting in year 2000 (SEN-C2-RCP45, SEN-C2-RCP85) are adjusted for the year 2000 using the REF-C2 reference.

We compare the CCM simulations with selected observations to provide a basic evaluation of their performance. In particular, to test the height-resolved evolution of the modelled ozone fields we use BSVertOzone v1.0 (Bodeker Scientific Vertical Ozone, hereafter referred to as "BSVertOzone"), which is an updated and further developed version of the BDBP (Binary Database of Profiles) v1.1.0.6 that is described in detail in *Bodeker et al.* (2013). BSVertOzone consists of monthly mean zonal mean ozone values on either pressure or altitude levels (from Earth's surface to about 70 km), where ozone is provided in mixing ratio or number density. For the data presented here the following improvements over the latest version of the BDBP were made: (1) ozone measurements from different data sources that are used as input for the monthly mean zonal mean calculation were updated (different satellite measurements and ozone soundings); (2) additional data sources were added (Microwave Limb Sounder, MLS, ozone profiles and recent years of ozone soundings); (3) drifts and biases between measurements from different data sources are adjusted (using a chemical transport model as transfer standard); (4) uncertainties are propagated from individual measurements through all preparation and calculation procedure steps to the final product; and (5) the calculation of the monthly mean zonal mean values was updated to correctly take into account the variable measurement frequencies of the different available data sources. The methodology of filling data gaps to construct a globally filled database is an updated version of the method described in *Bodeker et al.* (2013) where a pre-filling processing step was added. A more detailed description of BSVertOzone can be found in *Hassler et al.* (2018).

BSVertOzone spans 70 pressure levels that are approximately 1 km apart (878.4 hPa to 0.046 hPa). For the calculation of the partial columns, ozone was interpolated to the exact boundaries of the partial columns from the two closest BSVertOzone pressure levels, and then ozone was integrated between the determined levels. The boundaries for the partial columns were defined as follows: tropospheric column (surface-tropopause), lower stratospheric column (tropopause – 10 hPa), upper stratospheric column (10 hPa and above; for BSVertOzone this means up to 0.046hPa). The tropopause pressure was defined as 100 hPa in the tropics (20ºS-20ºN), 150 hPa in the mid-latitudes (20º-60ºN/S), and 200 hPa in the polar regions (60º-90ºN/S). CCM partial columns were integrated between the same partial column boundaries, but directly from the CCM pressure levels. No additional interpolation of CCM ozone profiles or BSVertOzone profiles was performed.

## 4) Results

### 4.1 Adjustment of model results

**Figure 1** shows October mean total column ozone (TCO) from the REF-C2 simulations for the Antarctic to illustrate how the model simulations compare before and after adjustment to fit observations. (Other regions are shown in the Supplementary Material (SM) **Figures S1-S5**). **Figure 1a** shows the mean TCO time series from the individual models (with a 3-point boxcar smoothing if only 1 realisation is available) along with three estimates of the model mean. These

are the overall multi-model mean (MMM), the mean of the models which lie within 1σ of the MMM (MMM1S), and the median model (MedM). While all models show the characteristic behaviour of depletion followed by recovery, there is a large spread of around 150 DU between the model values at any time. This complicates the determination of the ozone return dates relative to a baseline. Therefore, **Figure 1b** shows the same model runs after bias-correcting the individual model values to the observations over the period 1980-1984 and applying a 10-point boxcar smoothing. This correction to the models also forces the model means to fit the observations during this time period. With the bias correction, the individual models clearly give a more coherent picture and therefore, throughout this paper we generally show just the adjusted model results (unless indicated otherwise). However, it is important to note that the values of the multi-model mean, which are used for our best estimate of ozone return dates, are similar between **Figures 1a** and **1b.** Evidently, given enough model simulations, this approach of adjusting the model time series does not significantly alter the best estimate but does provide a smaller, more meaningful range of uncertainty. In other words, positive and negative model biases appear to be equally represented in the CCMI ensemble, but removing them will allow comparison of model returns to the same common 1980 baseline. Concerning the different methods to estimate the model average, for the Antarctic October case shown here the results are very similar. The difference is in the estimated range of variability which is shown by the shading. The 10$^{th}$ and 90$^{th}$ percentiles of the median give the largest spread of around 100 DU. The 1σ variation of the MMM is around ±40 DU. As expected the 1σ variation of the MMM1S, which has removed the outlying models, is smaller and around ±25 DU in this case. The MMM1S has an advantage over the median approach for the scenarios where there are not many realisations, and hence the 10$^{th}$/90$^{th}$ percentiles cannot be derived robustly. Therefore, we use the MMM1S and the 1σ deviation (of the forecasts provided by the adjusted models) to determine our best estimate of return dates and associated uncertainty, respectively.

## 4.2 Column ozone return dates

**Figure 2** compares the MMM1S TCO from REF-C2 with the standard CCMI historical simulations REF-C1 and REF-C1SD, in which the models are nudged towards analysed meteorology. Results are shown for both the direct and adjusted comparisons for the Antarctic and Arctic. The top panels show that, as expected, the REF-C1SD simulations reproduce better the observed evolution of TCO, as well as capturing much of the observed interannual variability. In particular, in the Antarctic the REF-C1SD mean reproduces the observed increase and then decrease in ozone after the year 2000. This was also shown in *Hardiman et al.* (2017) who investigated the contributions of dynamics, transport and chemistry to these differences and is consistent with chemical transport model analyses in *Chipperfield et al.* (2015, 2017). For the Arctic it is noticeable that the REF-C1SD simulations give significantly lower TCO than the REF-C1 in the 1990s. This is related to the series of cold Arctic winters during this period when low TCO was caused through large chemical loss (e.g. 1994/95 and 1995/96) along with dynamical contributions (e.g. 1996/97). The mean of the free-running REF-C1 simulations do not capture these low-column-ozone events. In contrast, **Figure 2** shows that the REF-C2 mean gives a more climatological picture of gradually varying ozone. The REF-C1 and REF-C2 means for the hindcast period are consistent. Therefore, within the limitations of the comparison of the heavily smoothed lines, the choice of ocean / sea ice representation does not seem to affect the climatological TCO evolution in the regions shown, although as noted by *Zhang et al.* (2016) sea ice loss may affect zonally asymmetric TCO trends.

**Figure 3** shows TCO results of the REF-C2 MMM1S for five latitude bands and the near-global (60ºS-60ºN) values. Also shown are the means from the sensitivity simulations SEN-C2-fGHG

and SEN-C2-fODS, which isolate the impacts of GHG and ODS changes. In all cases the shading gives the MMM1S 1σ variability. The TCO return dates from REF-C2, and the estimated range of that based on the 1σ variability, are summarised in **Table 3** and plotted in **Figure 4** (left panel). Globally the models predict a return to 1980 TCO values in 2049, with a 1σ spread from 2043-
2055. Earlier return dates are predicted in the NH mid-latitudes (2032), NH polar region (2034), and SH mid-latitudes (2045). In contrast, the return date from the large depletion in the SH polar region is much later at 2060. The 1σ variability gives the smallest range of return dates in the Antarctic (11 years) and SH mid-latitudes (11 years). The corresponding ranges in the NH, where dynamical variability is larger, are 24 years in mid-latitudes and 18 years near the pole. In the
tropics, the MMM1S shows a return to 1980 values towards 2058, followed by a turnaround and further decline. Note that individual models differ in whether they predict a return to 1980 values or not, before predicting the decrease at the end of this century (see SM **Figure S4**). The MMM1S near-global (60ºS-60ºN) column ozone also shows a decline after about 2080, which is mostly due to a decline in the tropics, with a small contribution from NH mid-latitudes. Dynamical decreases
in tropical TCO due to increased upwelling would be expected to cause increases in TCO at mid-high latitudes. For these variations in near global TCO after 2080 small changes of 2-3 DU in the tropospheric column, especially in the NH, are important factors (see **Figure 9** below). The right panel of **Figure 4** and **Table 4** show the return dates of stratospheric column ozone (SCO, see below).

Our predictions of TCO return dates can be compared with previous estimates, particularly those used in past WMO Ozone Assessments. WMO (2011) used results from the CCMVal-2 experiments to derive ozone return dates and these are shown in **Table 3**. As no major update to CCMVal-2 had occurred, these same CCMVal-2 values were also used as the best estimates in the
subsequent WMO (2014) assessment. However, that assessment also showed results from a subset of four CCMVal-2 models, selected for their good representation of stratospheric circulation (Figures 2-21 and 3-15 from that assessment), and analysis of five CMIP5 models (Figure 2-23; *Eyring et al*., 2013). We have included an analysis of these CMIP5 simulations in **Table 3**. Note that the CCMVal-2 runs used the A1b GHG scenario while the additional runs shown in WMO
(2014) used RCP 6.0 or 4.5. This study represents the first comprehensive update of TCO return dates since CCMVal-2. Compared to those values our return dates (albeit for a different GHG scenario) are later by values ranging from 4 years in the Arctic, 10 years in the Antarctic to 10-11 years at mid-latitudes. Similar differences are seen in the tropics, where not all models show a return. Interestingly, the subset of four CCMVal-2 models that performed new experiments for
WMO (2014), based on RCP 6.0 or 4.5, also showed later return dates than CCMVal-2, more in accord with our simulations. Compared to the CMIP5 models, which also use the RCP 6.0 scenario, our return dates are also later. The difference is small in northern mid-latitudes but it is as large as 14 years in the Antarctic (2046 compared to 2060).

There are three major differences in the assumptions used for the CCMVal-2 runs compared to the CCMI simulations, all of which likely contribute to our present estimate of up to a 17 year later return of total column ozone to the 1980 level. First, future atmospheric $CO_2$ rose more rapidly in the A1b scenarios used for CCMVal-2 compared to the RCP 6.0 scenario used by CCMI (**Figure S13**). Within A1b, $CO_2$ reaches 454 ppm in year 2032, which was the year given for
recovery of total column ozone in WMO (2011). This same mixing ratio for $CO_2$ is reached 9 years later within RCP 6.0. As a result, climate-driven changes in circulation within CCMI, which accelerate the recovery of the global ozone layer, will be similarly delayed. Next, the specification of $CH_4$ within A1b resulted in a more rapid rise compared to RCP 6.0 (**Figure S13**). For A1b in year 2032, atmospheric $CH_4$ is projected to be 44% larger than the 1980 value. For RCP 6.0, $CH_4$
is projected to rise by only 17% relative to 1980. As detailed in Section 4.5, the slower rise in $CH_4$ assumed for CCMI reduces the chemically-induced increase of ozone in both the stratosphere and

troposphere, contributing to a longer time for total column ozone to return to 1980 levels. The differences in the assumptions for future $N_2O$ between A1b and RCP 6.0 are small, and therefore not responsible for our later return dates. Finally, the return of effective equivalent stratospheric chlorine to 1980 levels is delayed by about 5 years in the baseline WMO (2011) scenario (used in this study) compared to the scenario used to drive CCMVal-2, reflecting new knowledge of lifetimes and emissions of various ODSs between the times these two scenarios were formulated. Since the return dates used in our study are based on analysis of results from 20 models, whereas the WMO (2011) return dates were based on results of 17 models, we have also examined the impact of restricting our return date estimates to a subset of the common models used by WMO (2011). Return dates for global and near-global ozone found using only 9 common models differ by only 1 to 3 years compared to values given in **Table 3** for the complete set of models. Therefore, the primary causes of the later dates for global and near-global ozone to return to 1980 levels reported in **Table 3**, compared to WMO (2011) return dates, are assumptions regarding future atmospheric levels of $CO_2$, $CH_4$, and ODSs inherent to CCMVAL-2 and CCMI. As our estimates are based on a large number of dedicated stratospheric simulations, with many models which have benefitted from further testing and development since CCMVal-2, we would argue that our values provide the best current estimates.

The sensitivity simulations in **Figure 3** also illustrate the effect of climate change and ODS changes on total column ozone recovery, confirming the results of previous studies (e.g. *Morgenstern et al.*, 2018 and references therein). The smallest impact of climate change is predicted in the Antarctic where the simulation with fixed GHGs is very similar to REF-C2. In the mid-latitudes, increasing GHGs brings forward the return dates by about 10 (NH) to 20 (SH) years. In the Arctic, there is a larger impact. However, the variation of the MMM1S line for SEN-C2-fGHG stays close to the 1980 reference line without crossing it, showing the challenge in extracting a return date (or range of dates) from the model runs. In the tropics, the competing effects of changes in ODSs, which increase ozone, and changes in GHGs, which decrease ozone, are clearly illustrated. In this region, the small rise in ozone due to ODS recovery is masked by decreases due to GHG increases. The net result is a decrease in tropical TCO in REF-C2.

### 4.3 Ozone profile variations

We now consider ozone recovery of partial columns in the lower (tropopause-10hPa; **Figure 5**) and upper (10 hPa and above; **Figure 6**) stratosphere separately. The figures include partial column observations derived from the composite BSVertOzone data set (*Bodeker et al.*, 2013; *Hassler et al.*, 2018). This composite dataset has the advantage, compared to observations from any single instrument, of being fully height resolved and available continuously over 1979-2016. BSVertOzone consists of several different data sets. We use here the Tier 1.4 data set that is based on observations, but has been created by applying a least squares regression model to attribute variability to various known forcing factors (natural and anthropogenic) for ozone. The variability in this data set is reduced compared to pure observations, since it describes only the variability for which basis functions were included in the regression model. This data set is therefore optimised for the use in comparisons with CCM simulations that do not exhibit the same unforced variability as reality. **Figures S10** and **S11** in the Supplementary Material show results from the specific altitudes of 50 hPa and 5 hPa, respectively, compared to GOZCARDS observations (*Froidevaux et al.,* 2015). In the lower stratosphere, where ozone has a long photochemical lifetime, the adjusted results from the models show some variations, especially in the polar regions. Overall, in the extra-tropical regions the models follow the observed behaviour in the BSVertOzone dataset although the MMM1S appears to overestimate depletion in the Antarctic and underestimate it in the Arctic. Interestingly, in the tropics the BSVertOzone dataset indicates ongoing decreases after

the year 2000 while the models show a levelling off and turnaround. This observed decrease in tropical lower stratospheric ozone has been noted by *Ball et al.* (2018) and is not captured by the models shown here. There are also significant differences between the models in the tropics where the lower stratosphere column does not return to 1980 values in all REF-C2 simulations. This is also the case for the near-global lower stratospheric column, reflecting the large influence of the tropics. In the upper stratosphere (**Figure 6**), ozone behaves more similarly in all regions and between all models (i.e. depletion followed by recovery to values larger than 1980) as dynamical variations are less important. Therefore, there is generally less spread in the model forecasts of ozone recovery than over the whole stratosphere, although the ULAQ-CCM, CHASER and CESM1-CAM4 models are outliers in certain regions. Note that the models consistently underestimate the BSVertOzone values and appear to show larger depletion. At this altitude, the feedback of temperature changes on ozone becomes important (*Haigh and Pyle*, 1982) and there is a larger increase in ozone than determined by ODS changes alone due to stratospheric cooling. As the CCMs are based on similar photochemical data, they should be expected to exhibit similar climate sensitivities, although they may predict different magnitudes of climate change. The reason for the disagreement between models and BSVertOZone in the upper stratosphere requires further investigation, but these upper stratospheric differences will not have a large impact on column ozone return years.

Since the main driver of past ozone depletion and ozone recovery is the evolution of stratospheric halogen loading, we pay particular attention to how well the CCMs model the time-dependent abundance of organic and inorganic chlorine and bromine. These are based on prescribed scenarios should be fields for which the CCMs agree well. **Figure 7** shows the evolution of the modelled inorganic chlorine ($Cly = HCl + ClONO_2 + HOCl + ClO + 2Cl_2O_2 + Cl + BrCl + OClO + …$) and total chlorine (Cly + organic) in the upper and lower stratosphere in the Antarctic from the REF-C2 simulations. This region was chosen as an example to illustrate model-model differences. Long-term variations in the modelled chlorine loading are determined by the specified surface mixing ratios of the ODS. Through this, the different models are constrained to show the same approximate timescale for chlorine to return to its values in e.g. 1980. If the stratospheric ODSs were simulated using emissions, rather than specified surface mixing ratios, then the timing of the model return dates would vary depending on the speed of the model circulation (*Douglass et al.*, 2008). Despite this constraint, the models do show a variation in the Cly loading at any time, especially in the lower stratosphere where the fractional conversion of organic chlorine to Cly is smaller. Nevertheless, some models show lower Cly than would be expected based on differences in circulation (EMAC-L47MA, EMAC-L90MA, CCSRNIES). Moreover, in the upper stratosphere, where most chlorine would be in the form of Cly, there are still large variations between the models. The reasons for this require further investigation beyond the scope of this study, but, if models are conserving chlorine in their chemistry and transport schemes, the evolution of Cly at 5 hPa should closely track that of the specified tropospheric chlorine with a lag of around 3-6 years. **Figure 7** also includes Microwave Limb Sounder (MLS) observations of the October vortex-mean volume mixing ratio of HCl + ClO. This provides a lower limit of the amount of Cly present, though it will be a good approximation in these locations. The year-to-year variation at 50 hPa (around 250 pptv) is due to variability in the polar vortex and chlorine activation, which is not an issue at 5 hPa. The comparison with the MLS data clearly shows that the three models with low Cly are unrealistic.

Bromine loading is another driver of ozone depletion and recovery. **Figure S12** compares total inorganic bromine modeled at 5 and 50 hPa in the Antarctic from the models. The highest values of bromine are reported by the SOCOL3 and EMAC-L90MA models. For SOCOL3, the model was run using surface mixing ratios of 1.63 and 1.21 ppt for $CHBr_3$ and $CH_2Br_2$, respectively, which results in a bromine content for these very short lived substances (VSLS) of 7.31 ppt. This

is larger than the 5 ppt estimate for total bromine of VSLS suggested for use in CCM models by *Eyring et al.* (2013a). Within EMAC-L90MA, bromine sources are explicitly represented by considering oceanic emission of $CHBr_3$ and $CH_2Br_2$, three other VSLS species, as well as sea salt (*Warwick et al.*, 2006). The tendency of Bry within EMAC-L90MA to be larger than found by other models, and to exceed an empirical estimate based on field observations in the Tropical Western Pacific, has been noted in a recent model intercomparison study focused on bromine (*Wales et al.*, 2018). The EMAC-L90MA over-estimate is likely caused by the oceanic emission terms being too large. The outlier is CHASER-MIROC, which has a bromine loading that peaks near 9 ppt in the upper stratosphere of the Antarctic, which is more than a factor of two less than surface mixing ratios of bromocarbons that reach the stratosphere (e.g., *Wales et al.*, 2018) as well as the mean value of peak Bry from the other models. The low value of Bry within CHASER-MIROC is due to the consideration of only $CH_3Br$, without scaling to represent either halons and VSLS bromocarbons, for the simulations conducted for CCMI. Within this model, the BrO+ClO cycle will have an unrealistically low effect on ozone.

The decline in stratospheric halogen loading is a main driver in the increase of stratospheric ozone. However, ozone is also affected by other stratospheric factors, notably driven by changing climate, such as rising temperature in the upper stratosphere. Moreover, differences in this estimated climate effect is a source of variations between the projections provided by the models analysed here. This climate effect is illustrated by the difference in the return dates to 1980 levels of stratospheric chlorine versus ozone. **Figure 8** (top and middle rows) shows how these return dates compare for local changes in the polar upper stratosphere and lower stratosphere, as well as column ozone return versus lower stratospheric Cly. For the upper stratosphere in both polar regions, the ozone return dates are much earlier than the Cly return dates and span a slightly larger range. In the lower stratosphere and for the column in the Antarctic there is a much closer correspondence between the Cly return date (at 50 hPa) and the ozone return date. For the Arctic, the ozone return dates are earlier than for Cly (see also **Figure 3**). **Figure 8** also shows the large spread in return dates between individual models. However, these return dates do generally correlate with each other so that an earlier Cly return date corresponds to an earlier ozone return date. Some exceptions to this occur – e.g. for the CMAM and EMAC-L90 models.

**Figure 9** shows the evolution of the tropospheric partial column ozone (PCO) from simulations based on scenario REF-C2. In the Antarctic (October), the MMM1S shows very little change from year 1980 through the forecast period. This is most likely due to limited tropospheric ozone precursor abundance in the polar SH region (e.g., low $NO_x$). In the other five regions, the MMM1S evolution typically shows 2-5 DU enhancement going from year 1980 to the peak tropospheric PCO in the forecast period. The models then generally show a decrease in the final few decades of the century. It is also interesting to note the large spread across participating CCMs in tropospheric PCO in the 21st century. For example, the SH and NH mid-latitude panels show, for year 2060, the range across the models is ~8-10 DU. The range is even larger near the end-of-the 21st century where the NH mid-latitudes and Arctic (March) ranges are ~15 DU and ~20 DU, respectively. Several models have less tropospheric PCO in year 2100 relative to year 1980. This is suggestive that the tropospheric chemistry sensitivity to mainly $CH_4$ is very different in the CCMs. There is more discussion of this topic in **Section 4.5**.

### 4.4 Sensitivity of ozone return to climate change

**Figure 10** shows the sensitivity of the tropospheric PCO evolution to assumed GHG RCP scenarios. The $CH_4$ temporal trend is the largest in SEN-C2-RCP85, followed by REF-C2, and SEN-C2-RCP45 scenarios. As discussed later in **Section 4.5**, an increase in tropospheric $CH_4$ will

enhance the $NO_x$-$HO_x$-smog net ozone production (*Haagen-Smit et al.,* 1950). Therefore, the larger future trend in $CH_4$ as represented in the SEN-C2-RCP85 scenario, relative to the REF-C2 scenario (**Figure S12**), increases the year 2100 global annual average MMM1S tropospheric PCO by ~5 DU. Previous studies (e.g. *Shindell et al.,* 2009; *Eyring et al.*, 2013b; *Morgenstern et al.*, 2018) have also found that the tropospheric impact of $CH_4$ increases is a major contributor to the TCO changes. Because of this difference in tropospheric PCO between scenarios we will only consider the evolution of SCO for derivation of return dates in the RCPs and SEN-C2 simulations in the discussion below.

**Figure 11** shows the sensitivity of the SCO return dates to the assumed GHG RCP scenarios. The relative change is smaller in the Antarctic, where recovery is largely determined by Cly loading, but larger in all other regions. However, the absolute changes between, for example, the Antarctic (October) and Arctic (March) are similar. Simulation SEN-C2-RCP26 (not shown), which assumes only small climate change but for which we only have 5 realisations, does not return to 1980 values at all except in the Arctic. Globally and at mid-low latitudes, simulation SEN-C2-RCP45 shows a behaviour between the RCP 2.6 and RCP 6.0 simulations. Compared to REF-C2, the simulation with the largest impact of climate change, SEN-C2-RCP85, shows a similar behaviour globally, but with regional differences, i.e. a positive effect on ozone at all latitudes except the tropics, where ozone decreases the most under RCP 8.5. The impact on return dates in different regions is summarized in **Figure 12**. This figure shows the return date for each model (coloured dots) along with the MMM1S (red triangle) for each scenario, all for six regions. The grey triangle is the MMM1S for the REF-C2 scenario (see **Figure 4,** right panel). The uncertainty in the MMM1S is represented by the solid vertical line. The SCO magnitude and range for all scenarios is listed in **Table 4**. For all regions shown in **Figure 12**, the derived MMM1S return date for SEN-C2-RCP45 is within the uncertainties range of the MMM1S return date for REF-C2. This is also true for SEN-C2-RCP85, although in the Antarctic (Oct.) region where the MMM1S return date for SEN-C2-RCP85 MMM1S is shortened by 12 years the uncertainly range only just overlaps with the MMM1S from REF-C2.

**4.5 Sensitivity of ozone return to methane and nitrous oxide**

We now focus on the sensitivity simulations (**Table 1**) which examine the individual roles of $CH_4$ (SEN-C2-fCH4 and SEN-C2-CH4RCP85) and $N_2O$ (SEN-C2-fN2O), and their combined impact with $CO_2$ (SEN-fGHG), on ozone recovery. We will only consider stratospheric ozone columns (SCO), thereby eliminating any impact from changes in tropospheric ozone discussed above. We first give a general overview based on prior studies of the expected ozone recovery impacts of changing $CH_4$, $N_2O$ and the combined GHGs. We then discuss the individual ozone recovery impacts of $N_2O$, $CH_4$, and GHGs based on this work. The derived impacts are compared to the REF-C2 recovery dates. We will not attempt here to diagnose the reason why models vary in the derived impacts; these details are beyond the scope of this study and will be addressed in future work.

Many studies have investigated the impacts of $CH_4$, $N_2O$, and $CO_2$ evolution on ozone abundance and recovery (e.g., *Haigh & Pyle*, 1979; *Le Texier et al.*, 1988; *Rosenfield et al.*, 2002; *Randeniya et al.*, 2002; *Royer et al.*, 2002; *Chipperfield and Feng*, 2003; *Portmann and Solomon*, 2007; *Shepherd et al.*, 2008; *Ravishankara et al.*, 2009; *Oman et al.,* 2010; *Fleming et al.*, 2011; *Revell et al.* 2012, 2015, 2016; *Kirner et al.*, 2015; *Butler et al.*, 2016; *Keeble et al.*, 2017). In summary, increases in $CH_4$ and $N_2O$ will generate higher amounts of hydrogen oxides ($HO_X$) and nitrogen oxides ($NO_X$), respectively. It is well known that increased $NO_X$ will enhance catalytic stratospheric ozone loss (*Crutzen*, 1970). Therefore, one would expect the ozone return date to be

extended for temporal increases in $N_2O$, or shortened for decreases in $N_2O$. Nonetheless, the effect of future increases of $N_2O$ varies with altitude and also depends on the temporal evolution of other GHGs (*Wang et al.*, 2014; *Revell et al.*, 2015). For changes in $CH_4$ the situation is more complicated. In a similar manner to $NO_X$, increased $HO_X$ will decrease upper stratospheric ozone.

However, $CH_4$ can also affect the partitioning of reactive chlorine through the reaction of $CH_4 + Cl \rightarrow HCl + CH_3$, with more $CH_4$ leading to an increase in stratospheric ozone via a decrease in the abundance of reactive chlorine. Overall, temporal increases in $CH_4$ lead to increases in stratospheric column ozone (*Revell et al.*, 2012). In the troposphere, increases in $CH_4$ will enhance chemical production through $NO_X$-$HO_X$-smog process (*Haagen-Smit et al.,* 1950). This

tropospheric ozone net production and subsequent tropospheric partial column change is shown for the REF-C2 simulations in **Figure 9**. It should be noted that UMSLIMCAT shows a small tropospheric trend since the ozone is prescribed in the troposphere. UMUKCA-UCAM has only simplified tropospheric chemistry, whereas NIWA-UKCA has a representation of $C_2$-$C_3$-isoprene oxidation. In addition, the tropospheric trend is affected by the coupling to the stratosphere via

changes in stratotosphere-to-troposphere exchange and photolytic feedbacks. Although $CO_2$ is chemically inert below about 60 km, increases in its abundance (along with $CH_4$ and $N_2O$) will cool the stratosphere (*Haigh & Pyle*, 1979). This cooling will slow down the catalytic ozone destruction cycles and increase ozone, therefore temporal increases in $CO_2$, $CH_4$, and $N_2O$ will shorten the ozone return date due to this process, which is most important in the upper stratosphere

and mesosphere. The warming of the troposphere and the cooling of the stratosphere can also affect the Brewer-Dobson circulation and therefore impact ozone through transport (e.g., *Polvani et al.*, 2017, 2018 and references therein). The cooling process operates throughout the stratosphere, but is most important for dynamical processes in the lower to middle stratosphere. Outside of the tropics, a speed-up of the Brewer-Dobson circulation would shorten the ozone

recovery date, while a slow-down of the BD circulation would extend it.

All of the above-mentioned chemical, radiative, and dynamical impacts are represented within the REF-C2 simulations (using RCP 6.0 GHGs for the future period). Here we examine the sensitivity scenarios for $N_2O$ and $CH_4$ individually, along with the combined GHGs scenario impacts relative

to the REF-C2 scenario. It should be noted that for the two $CH_4$ and one $N_2O$ sensitivity scenarios, there is only one realisation available for each model, whereas for many models the REF-C2 scenario has multiple ensemble members (**Table 2**). The SEN-C2 temporal abundances compared to REF-C2 are shown for $N_2O$, $CH_4$, and $CO_2$ in **Figure S13.** For $N_2O$ (SEN-C2-fN2O), the abundance is approximately 290 ppbv and 405 ppbv for 1960 and 2100, respectively, an increase

of 115 ppbv (~40%). For $CH_4$ (SEN-C2-fCH4) the abundance is 1.24 ppmv in 1960 and a maximum of 1.96 ppmv in the 2070s, an increase of 57%. The SEN-C2-RCP85 scenario increases $CH_4$ over that given by the REF-C2 scenario by 2.1 ppmv in 2100, an increase of 128%. The $CO_2$ change in REF-C2 from 1960 to 2100 is 352 ppmv, approximately a 110% increase.

**Figure 13** shows the evolution of October mean SCO in the Antarctic region for the REF-C2, SEN-C2-fN2O (fixed $N_2O$ at 1960 conditions), SEN-C2-fCH4 (fixed $CH_4$ at 1960 conditions), and SEN-C2-CH4RCP85 (RCP8.5 $CH_4$ abundance) scenarios. The SCO observations are again based on the BSVertOzone data set (*Bodeker et al.*, 2013). The panels show results from eight different models. Solid lines show 10-year smoothed SCO for a given simulation. Shading

indicates the 1-σ standard derivation from an ensemble of realisations (or 3-box smoothed line if only one realisation is available). In addition, the SEN-C2-fODS (fixed ozone-depleting substances in 1960) is shown. This scenario shows the behaviour of SCO without the evolution of halogens (i.e., no ozone depletion and recovery due to halogens). This scenario does include the previously discussed impacts of $N_2O$, $CH_4$, and $CO_2$ on SCO. However, since there is no ozone

depletion period from ODSs in this simulation, it does not make sense to calculate an ozone recovery date. The red lines in all panels show the evolution of SCO for the REF-C2 scenario,

which can be directly compared to the BSVertOzone dataset. For each model, the REF-C2 simulations have been bias corrected to the BSVertOzone dataset for the 1980-1984 period. The SEN-C2 simulations are then adjusted to the bias corrected REF-C2 ensemble mean for the 1960 period. Comparison of the SCO for the REF-C2 simulations show that models generally compare

well to BSVertOzone for the depletion period, although CMAM and UMSLIMCAT appear to be biased low. In summary, when one examines the relative impact on the ozone return date across the eight models from the four SEN-C2 scenarios, there is no consistent pattern. Therefore, the result suggests that the Antarctic region is not sensitive to the perturbations presented in this work.

This is not the case when one examines the annual average SH mid-latitude region (**Figure 14)**. Generally, across the models the ozone return date varies as follows. The SEN-C2-fN2O (light blue line) simulations return before the REF-C2 simulations. This is consistent with our understanding that less $NO_X$ produced from a fixed 1960 $N_2O$ abundance will allow the SCO to recover earlier than the increasing $NO_X$ in a REF-C2 scenario. However, it should be noted that

there can be a smaller impact of $N_2O$ on the return date due to cancellation of the upper and lower stratospheric response of $N_2O$ on ozone (*Morgenstern et al.*, 2018). The SCO from the SEN-C2-CH4RCP85 simulations (dark blue line), for the four models which performed this run, also tends to have a recovery date that is earlier than the REF-C2 simulations. Again, with more $CH_4$ specified in the 21$^{st}$ century, ozone will recover faster due to the sequestering of reactive chlorine

into HCl and the stratospheric cooling effect of slowing down ozone loss rates. The impact of increased $HO_X$ production from increased $CH_4$, causing more ozone depletion to extend the recovery, does not dominate over these two processes. In contrast, the SEN-C2-fCH4 simulation (purple line), has less $CH_4$ in the 21$^{st}$ century than the REF-C2 simulation and therefore has later return date. Finally, the SEN-C2-fGHG simulation generally has the latest ozone return date. The

corresponding time evolution results for the SEN-C2 scenarios for the Arctic (March), annual average NH mid-latitude region, and annual average near-global (60°S-60°N) are shown in **Figures S14, S15**, and **S16**, respectively.

The SCO return dates for the simulations based on the four sensitivity scenarios are also

summarised and compared to the REF-C2 scenario in **Figure 12** (see above) and **Table 4**. The individual model details regarding the SCO return date (similar in format to **Figure 8**) between the SEN-C2 simulations and the REF-C2 simulations are shown in **Figures S17-S20**. We first discuss the change in SCO return dates between the SEN-C2-fGHG and REF-C2 simulations. The Antarctic (October) region difference between the two scenarios is small, within 2 years. The

uncertainty range for both scenarios are approximately ±12 years. The SH mid-latitudes region shows that the MMM1S SCO recovery date is extended in the SEN-C2-fGHG case by ~16 years relative to the REF-C2 case. This extended SCO recovery period is even larger in the NH mid-latitudes and Arctic (March) by ~25 years. This is consistent with the GHG cooling impact on ozone loss rates and a lack of strengthening of the BD circulation. In this comparison, having 1960

abundances of GHGs compared to the REF-C2 evolution means less cooling of the stratosphere and therefore an extension of the SCO recovery date. In addition, the $CH_4$ abundance is less in SEN-C2-fGHG which also decreases sequestering of reactive chlorine into HCl and acts to extend the SCO recovery date. Both of these factors override the direct impact of less production of $NO_X$ and $HO_X$ from $N_2O$ and $CH_4$ which would shorten the SCO recovery. Interestingly the near-global

(annual) average SCO return date is not that different between the two scenarios. This is most likely due to the fact that when the BD circulation strengthens, tropical ozone is reduced and extratropical ozone is increased. Therefore, the net impact on the stratospheric column ozone return date cancels out for this process in the global average. **Figure 12** and **Table 4** shows the comparison of the SCO return dates for the SEN-C2-fN2O and the REF-C2 simulations. In this

comparison, one would expect that SEN-C2-fN2O with 1960 abundances of $N_2O$ would bring forward the SCO recovery date. This is certainly true for the near-global (annual) average

comparison, where the MMM1S SEN-C2-fN2O SCO recovery date is shortened by ~20 years relative to the REF-C2 case. This is mostly due to a shortening of the return date in the tropics; at mid-high latitudes there is little change. As noted above, the future rise in $N_2O$ can lead to significant increases in lower stratospheric ozone, particularly for regions where the loss rate of

5 ozone due to halogens exceeds that due to $NO_x$ prior to the perturbation of $N_2O$. The effect of $N_2O$ on ozone varies as a function of latitude and altitude (*Wang et al.*, 2014), complicating the sensitivity to the ozone return date to variations in $N_2O$ (*Morgenstern et al.,* 2018). **Figure 12** and **Table 4** also shows the comparison of the SCO return dates for the SEN-C2-fCH4 simulations. Here all regions except the Antarctic (and the tropics, for which the return date is undefined) show

an extension of the SCO return date. This is consistent with the discussion above for the fixed GHG scenario. The near-global (annual) average SCO return date is extended by ~5 years. For the SEN-C2-CH4RCP85 scenario the MMM1S near-global (annual) average SCO return date is reduced relative to the REF-C2 scenario by ~15 years. As expected, this is an opposite effect to that derived from the SEN-C2-fCH4 scenario. In the other regions for SEN-C2-CH4RCP85

MMM1S there is less separation between the REF-C2 reference and the $CH_4$ perturbation scenario. Unfortunately for this work, there was only one realisation from each modelling group for the SEN-C2-fN2O, SEN-C2-fCH4, and SEN-C2-CH4RCP85 scenarios. Therefore, the signal from the perturbation on the SCO return date may be affected by the internal variability of each CCM. For future assessments of single forcing or combined effects of $CO_2$, $CH_4$, and $N_2O$ on SCO

the return date, a recommendation based on this work is to have at least three ensemble members per scenario type.

## 5) Summary and Conclusions

We have analysed simulations performed for the Chemistry-Climate Modelling Initiative (CCMI)

to estimate the return dates of the stratospheric ozone layer from depletion caused by anthropogenic stratospheric chlorine and bromine. CCMI represents an extensive multi-model exercise to study the future evolution of the ozone layer under changing climate conditions. Here we consider a total of 155 simulations from 20 models, including a range of sensitivity studies. For the control simulations (unconstrained by nudging towards analysed meteorology) there is a

large spread in the predictions of the absolute ozone column. Therefore, the model results need to be adjusted for biases against historical data. Also, the interannual variability in the model results need to be smoothed in order to provide a reasonable and useful estimate of the range of ozone return dates.

The total column ozone return dates calculated here differ from those presented in the previous WMO/UNEP assessment (WMO, 2014). The differences could be explained by the choice of GHG scenario for the baseline estimate (A1b in WMO (2014) versus RCP 6.0 here), and some model updates including realistic tropospheric chemistry. In addition, the time series for surface ODSs used here results in a return of stratospheric halogen loading to the 1980 value that is about

5 years later than assumed for the CCM runs examined in WMO (2014). The CCMI models project that global total column ozone will return to 1980 values in 2049 (with a 1-σ uncertainty of 2043-2055). At mid-latitudes, Southern Hemisphere ozone is projected to return to 1980 values in 2045 (2039-2050), and Northern Hemisphere ozone in 2032 (2020-2044). In the polar regions, the return dates are 2060 (2055-2066) in the Antarctic in October and 2034 (2025-2043) in the Arctic

in March. The earlier return dates in the NH reflect larger impact of dynamics on ozone in this hemisphere. In the tropics only around half the models predict a return to 1980 values, at around 2040, while the other half do not show a return to 1980 values (giving the mean of 2058). All models show a negative trend in tropical total column ozone towards the end of the 21$^{st}$ century.

An important result from the simulations presented here is the strong regional differences in the future evolution of total column ozone due to the effects of climate change. These climate effects are least evident in the Antarctic spring where future ozone depends largely on halogen loading. In contrast, in the NH the models predict a super-recovery while in the tropics the models predict a possible further decrease in column ozone, possibly without any return to 1980 values.

There is different behaviour in the partial column ozone between the lower and upper stratosphere. In the lower stratosphere, where ozone is long-lived and affected by dynamics, there are differences in the timescale for recovery between the polar regions and mid-latitudes. Moreover, in the tropics, increased upwelling prevents the return of partial column ozone (PCO) in many models. In contrast in the upper stratosphere the predicted behaviour is similar in all regions. Ozone returns to values larger than in 1980 by 2040 and carries on increasing due to the effect of stratospheric cooling. For the upper stratosphere, the CCM predictions do not vary a lot and are in good agreement with past observations, indicating that relevant processes are represented adequately in the models. For the lower stratosphere, some obvious differences are seen between the CCM results, indicating possible inadequate descriptions of dynamical (transport) and chemical (heterogeneous) processes due to temperature biases (in the polar regions and tropics) in the CCMs.

The CCMI models generally agree in their simulation of the time evolution of stratospheric inorganic chlorine (Cly), which is the main driver of ozone loss and recovery, although there is some inter-model variability. The situation is similar for the simulation of inorganic bromine but with a larger relative spread between the models due to varying treatments of short-lived species and a model without a representation of halons. However, there do appear to be issues with the chemistry and/or transport in a few of the model simulations. Throughout the stratosphere the spread of ozone return dates to 1980 values between models tends to correlate with the spread of the return of Cly to 1980 values. In the upper stratosphere, greenhouse gas-induced cooling speeds up the return by about 10-20 years. In the lower stratosphere, and for the column, there is a more direct link in the timing of the return dates, especially for the large Antarctic depletion. Comparisons of total column ozone between the models is affected by different predictions of the evolution of tropospheric ozone within the same scenario, presumably due to differing treatment of tropospheric chemistry. Therefore, for some scenarios clear conclusions can only be drawn for stratospheric ozone columns.

As noted by previous studies, the timing of ozone recovery is affected by the evolution of $N_2O$ and $CH_4$. However, the effect in the simulations analysed here is small and at the limit of detectability from the small number of realisations available for these experiments compared to internal model variability. The sensitivity scenarios examined in this study did show that a future decline of $N_2O$ below an RCP6.0 projection (i.e., back to 1960 abundances) could reduce the global SCO return date by up to 15 years. In the opposite sense, an increase of $CH_4$ above the RCP6.0 projection (i.e., using RCP8.5 abundances) could again reduce the global SCO return date by up to 15 years. Both the $N_2O$ and $CH_4$ global column ozone sensitivities are mainly realised through chemical impacts in the tropics.

Overall, our estimates of ozone return dates are uncertain due to both uncertainties in future scenarios, in particular of GHGs, and uncertainties in models. The scenario uncertainty is small in the short term but increases with time, and is large by the end of the century. For the models, while it is possible that they all may be missing important but unknown processes, there are still some model-model differences related to known first-order processes which affect ozone recovery. Work needs to continue to ensure that models used for assessment purposes accurately represent stratospheric chemistry and the prescribed scenarios of ozone-depleting substances, and only those

models are used to calculate return dates. Nevertheless, the agreement between the results presented here and previously published work gives some confidence that we can model the future evolution of the ozone layer. For future assessments of single forcing or combined effects of $CO_2$, $CH_4$, and $N_2O$ on the stratospheric column ozone return dates, this work suggests that is more important to have multi-member (at least 3) ensembles for each scenario from each established participating model, rather than a large number of individual models.

**Acknowledgements**:
We acknowledge the modeling groups for making their simulations available for this analysis, the joint WCRP IGAC/SPARC Chemistry-Climate Model Initiative (CCMI) for organizing and coordinating the model data analysis activity, and the British Atmospheric Data Centre (BADC) for collecting and archiving the CCMI model output. We thank Michelle Santee for providing the MLS data. The SBUV Merged Ozone Data Set is made available under the NASA Long Term Measurement of Ozone program WBS 479717. O.M., G.Z., N.L.A., E.M.B. and J.A.P. acknowledge the UK Met Office for use of the MetUM. NIWA research was supported by the NZ Government's Strategic Science Investment Fund (SSIF) through the NIWA programme CACV. O.M. acknowledges funding by the New Zealand Royal Society Marsden Fund (grant 12-NIW-006) and by the Deep South National Science Challenge (www.deepsouthchallenge.co.nz). We acknowledge the contribution of NeSI high-performance computing facilities to the results of this research. New Zealand's national facilities are provided by the New Zealand eScience Infrastructure (NeSI) and funded jointly by NeSI's collaborator institutions and through the Ministry of Business, Innovation & Employment's Research Infrastructure programme (www.nesi.org.nz). N.L.A., A.T.A., E.M.B. and J.A.P. acknowledge use of the ARCHER UK National Supercomputing Service (www.archer.ac.uk) and the MONSooN system, a collaborative facility supplied under the Joint Weather and Climate Research Programme, which is a strategic partnership between the UK Met Office and NERC. E.M.B. acknowledges funding from the ERC for the ACCI project (grant number 267760). The SOCOL team acknowledges support from the Swiss National Science Foundation under grant agreement CRSII2_147659 (FUPSOL II). E.R. acknowledges support from the Swiss National Science Foundation under grant 200021_169241 (VEC). H.A. acknowledges Environment Research and Technology Development Fund of the Environmental Restoration and Conservation Agency, Japan (2-1303 and 2-1709) and NEC-SX9/A(ECO) computers at CGER, NIES. The IPSL team acknowledges support from the Centre d'Etude Spatiale (CNES) SOLSPEC grant, European Project StratoClim (7th Framework Programme, grant agreement 603557) and the LABEX L-IPSL, funded by the French Agence Nationale de la Recherche under the "Programme d'Investissements d'Avenir". N.B., S.C.H. and F.M.O'C. were supported by the Joint UK BEIS/Defra Met Office Hadley Centre Climate Programme (GA01101). N.B. and S.C.H. also acknowledge the European Commission's 7th Framework Programme, under grant agreement no. 603557, StratoClim project. F.M.O'C. acknowledges additional support from the Horizon 2020 European Union's Framework Programme for Research and Innovation Coordinated Research in Earth Systems and Climate: Experiments, kNowledge, Dissemination and Outreach (CRESCENDO) project under grant agreement no. 641816. The EMAC simulations have been performed at the German Climate Computing Centre (DKRZ) through support from the Bunesministerium für Bildung und Forschung (BMBF). DKRZ and its scientific steering committee are gratefully acknowledged for providing the HPC and data archiving resources for this consortial project ESCiMo (Earth System Chemistry integrated Modelling). CESM1 (WACCM) and CESM1 (CAM4) are components of NCAR's CESM, which is supported by the NSF and the Office of Science of the U.S. Department of Energy. Computing resources were provided by NCAR's Climate Simulation Laboratory, sponsored by NSF and other agencies. This research was enabled by the computational and storage resources of NCAR's Computational and Information Systems Laboratory (CISL). R.S. and K.S. acknowledge support from Australian Research Council's Centre of Excellence for Climate System Science (CE110001028), the Australian Government's National Computational Merit Allocation Scheme (q90) and Australian Antarctic science grant program (FoRCES 4012). S.K. acknowledges funding by the Deep South National Science Challenge (CO1X1445). R.J.S. appreciates support of the National Aeronautics and Space Administration ACMAP and Aura programs. M.P.C. and S.D. acknowledge use of the Archer and Leeds HPC facilities and funding from the NERC SISLAC project (NE/R001782/1). M.P.C. thanks the Royal Society for a Wolfson Merit Award.

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

## Tables

**Table 1.** Definitions of CCMI scenarios. Adapted from *Eyring et al.*, (2013a).

| Simulations | Period | GHGs | ODSs | SSTs / SICs | Bqckground & Volcanic Aerosols | Solar Variability | VSLS Bromine | QBO | Ozone and Aerosol precursors |
|---|---|---|---|---|---|---|---|---|---|
| REFERENCE SCENARIOS | | | | | | | | | |
| REF-C1 | 1960-2010 | OBS + CMIP5 updated to 2010 | OBS (WMO-2011) | OBS, HadlSST1 | OBS Sulfate SAD & radius | Spectrally resolved irradiance, Proton ionization, Ap | YES ~5 pptv additional Bry | OBS or internally generated | OBS Based on Lamarque, (2010). |
| REF-C1SD | 1980-2010 | REF-C1 | REF-C1 | Consistent with met fields. | REF-C1 | REF-C1 | REF-C1 | Consistent with met fields. | REF-C1 |
| REF-C2 | 1960-2100 | Obs + RCP6.0 | REF-C1 | Model SSTs | Obs and background for forecast period. | REF-C1, repeating SV in the future | REF-C1 | Consistent with REF-C1 approach | REF-C1 until + RCP6.0 scn in the future |
| RCP SENSITIVITY | | | | | | | | | |
| SEN-C2-RCP26 | 2000-2100 | RCP2.6 | REF-C2 | Model SSTs (RCP2.6) | REF-C2 | REF-C2 | REF-C2 | REF-C2 | Based on RCP2.6 in the future |
| SEN-C2-RCP45 | 2000-2100 | RCP4.5 | REF-C2 | Model SSTs (RCP4.5) | REF-C2 | REF-C2 | REF-C2 | REF-C2 | Based on RCP4.5 in the future |
| SEN-C2-RCP85 | 2000-2100 | RCP8.5 | REF-C2 | Model SSTs (RCP8.5) | REF-C2 | REF-C2 | REF-C2 | REF-C2 | Based on RCP8.5 in the future |
| ODS and GHG SENSITIVITY | | | | | | | | | |
| SEN-C2-fODS | 1960-2100 | REF-C2 | Fixed ODSs at 1960 | REF-C2 | REF-C2 | REF-C2 | REF-C2 | REF-C2 | REF-C2 |
| SEN-C2-fGHG | 1960-2100 | Fixed GHGs at 1960 | REF-C2 | REF-C2 | REF-C2 | REF-C2 | REF-C2 | REF-C2 | REF-C2 |
| SINGLE FORCING | | | | | | | | | |
| SEN-C2-fCH4 | 1960-2100 | REF-C2, w/Fixed $CH_4$ at 1960 | REF-C2 | REF-C2 | REF-C2 | REF-C2 | REF-C2 | REF-C2 | REF-C2 |
| SEN-C2-fN2O | 1960-2100 | REF-C2, w/Fixed $N_2O$ at 1960 | REF-C2 | REF-C2 | REF-C2 | REF-C2 | REF-C2 | REF-C2 | REF-C2 |
| SEN-C2-CH4RCP85 | 2000-2100 | REF-C2, w/$CH_4$ from RCP8.5 | REF-C2 | REF-C2 | REF-C2 | REF-C2 | REF-C2 | REF-C2 | REF-C2 |

**Table 2**. CCMI simulations analysed in this study. The numbers indicate the number of realisations by each model for each simulation.

| Model name | REF-C1 | REF-C1SD | REF-C2 | SEN-C2-fGHG | SEN-C2-fODS | SEN-C2-RCP26 | SEN-C2-RCP45 | SEN-C2-RCP85 | SEN-C2-fCH4 | SEN-C2-CH4RCP85 | SEN-C2-fN2O | Total Simulations |
|---|---|---|---|---|---|---|---|---|---|---|---|---|
| ACCESS CCM | 1 | | 2 | 2 | 2 | | | | | | | 7 |
| CCSRNIES MIROC3.2 | 3 | 1 | 2 | 1 | 1 | 1 | 1 | 1 | 1 | | 1 | 13 |
| CESM1 CAM4-CHEM | 3 | | 3 | | | | | | | | | 6 |
| CESM1 WACCM | 5 | 1 | 3 | 3 | 3 | | 1 | 3 | 1 | 1 | 1 | 22 |
| CHASER (MIROC-ESM) | | 1 | 1 | 1 | 1 | 1 | | 1 | 1 | | 1 | 8 |
| CMAM | 3 | 1 | 1 | 1 | 1 | 1 | 1 | 1 | 1 | 1 | 1 | 13 |
| CNRM-CM5-3 | 4 | 1 | 2 | | | | | | | | | 7 |
| EMAC-L47 | 1 | 1 | 1 | | | | | | | | | 3 |
| EMAC-L90 | 1 | 1 | 1 | | | | | | | | | 3 |
| GEOSCCM | 1 | | 1 | | | | | | 1 | 1 | 1 | 5 |
| GFDL-CM3/AM3 | 1 | 1 | 1 | | | | 3 | 1 | | | | 7 |
| HadGEM3-ES | 1 | 1 | 1 | | | | | | | | | 3 |
| IPSL-LMDZ-REPROBUS | 1 | 1 | 1 | | | | | | | | | 3 |
| MRI | 1 | 1 | 1 | | | | | | | | | 3 |
| NIWA-UKCA | 3 | | 5 | 3 | 2 | | | | 1 | | 1 | 15 |
| SOCOL3 | 3 | | 1 | | | 1 | 1 | 1 | 1 | 1 | 1 | 10 |
| TOMCAT (CTM) | 1 | 1 | | | | | | | | | | 2 |
| ULAQ CCM | 3 | | 3 | 1 | 1 | 1 | 1 | 1 | 1 | 1 | 1 | 14 |
| UMSLIMCAT | 1 | | 1 | 1 | 1 | | 1 | 1 | | 1 | | 7 |
| UMUKCA-UCAM | 1 | 1 | 2 | | | | | | | | | 4 |
| | | | | | | | | | | | | |
| Total realisations | 38 | 13 | 33 | 13 | 12 | 5 | 9 | 10 | 8 | 6 | 8 | 155 |
| Total models | 19 | 13 | 19 | 8 | 8 | 5 | 7 | 8 | 8 | 6 | 8 | |

**Table 3**. Total column ozone (TCO) return dates to 1980 baseline from REF-C2 simulations using different averaging methods. Values in brackets indicate recovery dates based on either 1-σ standard deviation or 10th and 90th percentile estimates. The number *2100* in italics indicates that the estimated ozone uncertainty range has not returned to the 1980 values within the time range of the model simulation. The MMM for RCP 6.0 derived from the CMIP5 models (*Erying et al.*, 2013b) is shown in column two.

| | WMO[1] (2011, 2014) | CMIP5[2] *Eyring et al.* (2013) | CCMI REF-C2 (this work) | | |
| --- | --- | --- | --- | --- | --- |
| | | | MMM | Median | MMM1S |
| SH pole (October) | 2050 (2045-2060) | 2046 (2040-2055) | 2062 (2051-2082) | 2061 (2042-2069) | 2060 (2055-2066) |
| SH Mid-latitudes | 2035 (2030-2040) | 2041 (2033-2046) | 2046 (2038-2053) | 2046 (2038-2071) | 2045 (2039-2050) |
| Tropics | 2042 (2028-     ) | N/A | *2100* (2034-*2100*) | 2058 (2013-*2100*) | 2058 (2038-*2100*) |
| NH Mid-latitudes | 2021 (2017-2026) | 2032 (2026-2039) | 2033 (2011-2047) | 2032 (2010-2048) | 2032 (2020-2044) |
| NH pole (March) | 2030 (2025-2035) | 2028 (2020-2033) | 2039 (2021-2050) | 2034 (2011-2058) | 2034 (2025-2043) |
| Near Global (60°S-60°N) | | 2043 (2035-2050) | 2045 (2034-2064) | 2046 (2025-*2100*) | 2047 (2042-2051) |
| Global | 2032 (2027-2038) | | 2046 (2035-2058) | 2048 (2040-2073) | 2049 (2043-2055) |

1. Based on CCMVal-2 model simulations (A1b GHG scenario) and reported in WMO (2011) and Table 2-5 and Figure 3-16 of WMO (2014).
2. Based on CMIP5 models used in Figure 2-23 of WMO (2014) with the point-wise 95% confidence interval. This approach to estimating uncertainties was also used in *Eyring et al.* (2013b), *Eyring et al.* (2010b) and Chapter 3 of WMO (2011).

**Table 4**. Stratospheric column ozone (SCO) return dates to 1980 baseline from various simulations. Values in brackets indicate the range of recovery dates based on 1-σ standard deviation. The number *2100* in italics indicates that the estimated ozone uncertainty range has not
5  returned to the 1980 values within the time range of the model simulation. Simulations starting in year 2000 (SEN-C2-CH4RCP85, SEN-C2-RCP45 and SEN-C2-RCP85) use 1980 baseline from REF-C2 simulations.

| | REF-C2 | SEN-C2-fGHG | SEN-C2-fN2O | SEN-C2-fCH4 | SEN-C2-CH4RCP85 | SEN-C2-RCP45 | SEN-C2-RCP85 |
|---|---|---|---|---|---|---|---|
| **SH pole (October)** | 2061 (2054-2067) | 2059 (2052-2064) | 2061 (2055-2066) | 2062 (2058-2068) | 2056 (2051-2062) | 2052 (2046-2060) | 2049 (2045-2055) |
| **SH Mid-latitudes** | 2045 (2039-2049) | 2061 (2055-2066) | 2043 (2037-2047) | 2054 (2050-2060) | 2042 (2036-2048) | 2043 (2036-2049) | 2042 (2036-2048) |
| **Tropics** | N/A | 2049 (2043-2068) | 2054 (2039-*2100*) | N/A | N/A | N/A | N/A |
| **NH Mid-latitudes** | 2035 (2028-2045) | 2060 (2055-2079) | 2026 (2022-2033) | 2043 (2039-2047) | 2034 (2027-2043) | 2035 (2029-2042) | 2032 (2027-2044) |
| **NH pole (March)** | 2031 (2025-2036) | 2058 (2040-2064) | 2030 (2011-2037) | 2037 (2027-2041) | 2030 (2022-2036) | 2028 (2022-2044) | 2027 (2023-2038) |
| **Near Global** | 2058 (2046-2077) | 2060 (2056-2075) | 2040 (2035-2047) | 2063 (2056-2088) | 2044 (2038-2050) | 2066 (2044-*2100*) | 2056 (2045-2080) |
| **Global** | 2049 (2045-2059) | 2058 (2052-2064) | 2042 (2035-2048) | 2057 (2051-2074) | 2043 (2037-2049) | 2055 (2044-2078) | 2047 (2042-2056) |

**Figures**

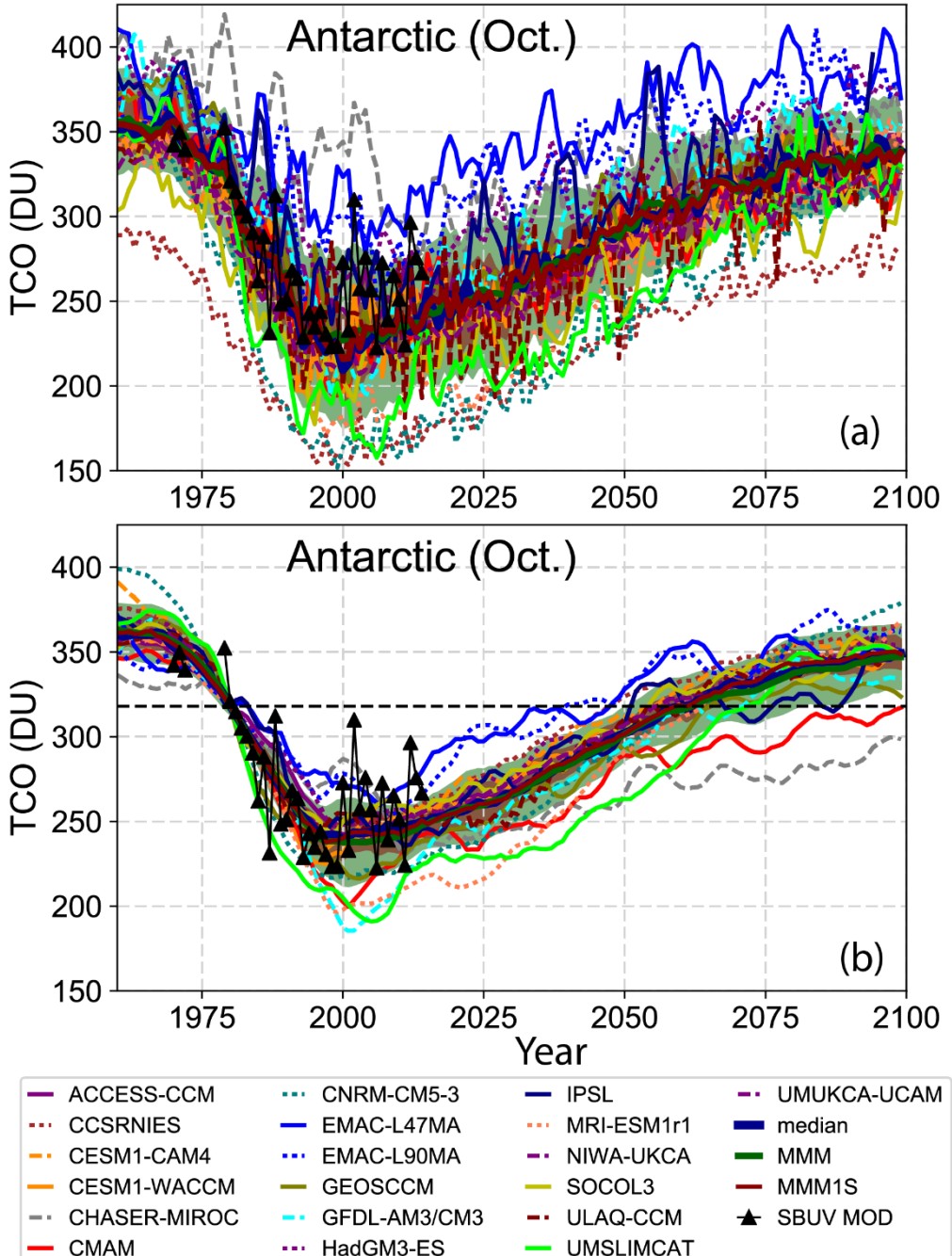

**Figure 1**. (a) Total column ozone time series (DU) for Antarctic in October from 19 individual
CCMs for the REF-C2 simulations along with observations from the Solar Backscatter
Ultraviolet (SBUV) merged ozone dataset (MOD) (*Frith et al.*, 2017). The MMM, median
(MedM) and MMM1S are shown with thick green, blue and red lines, respectively. The light
blue shaded region indicates the 10[th] and 90[th] percentile range. Light green and red regions show
1-σ variability w.r.t. MMM and MMM1S lines, respectively. (b) Same as panel (a) but adjusted
total ozone time series w.r.t. mean 1980-1984 observations and after application of 10-point
boxcar smoothing. The dashed black line indicates 1980 reference value.

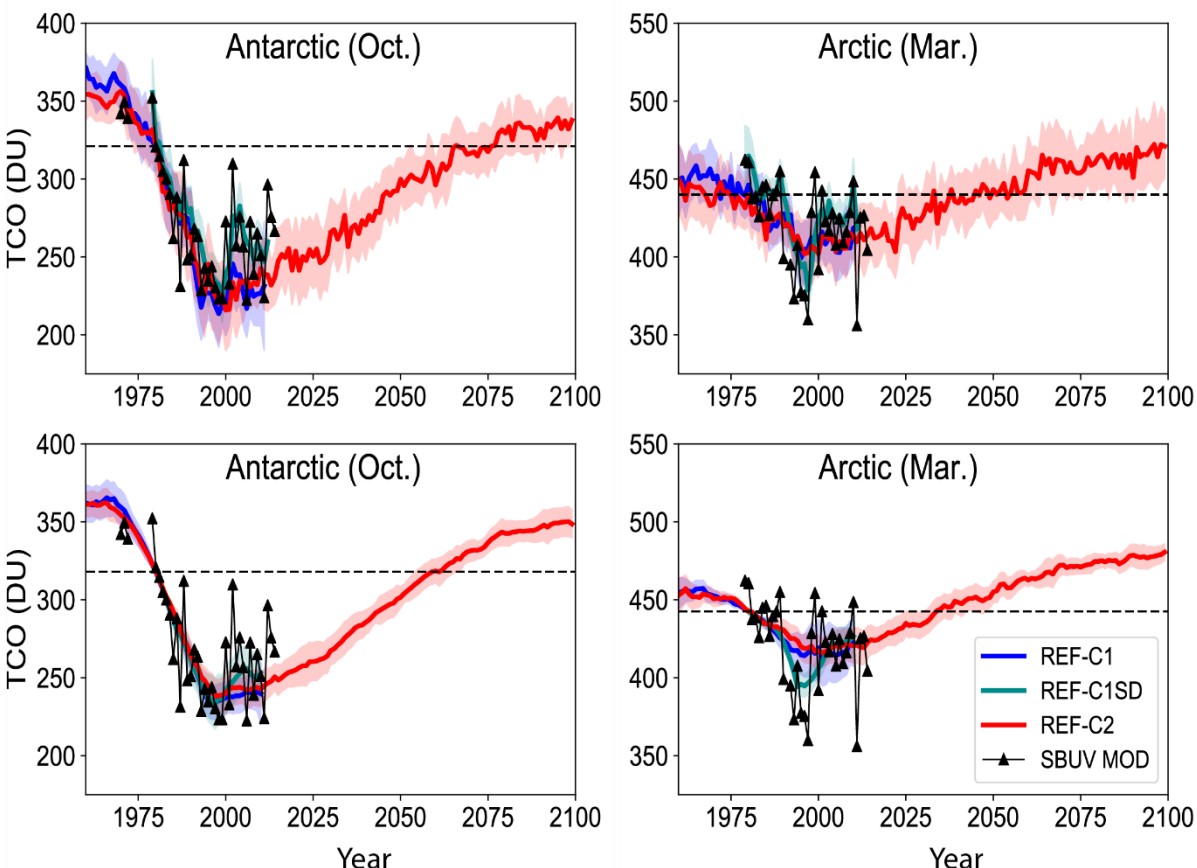

**Figure 2**. MMM1S total column ozone time series (DU) from REF-C1 (blue), REF-C1SD (dark cyan) and REF-C2 (red) simulations for the (left) SH polar (October) and (right) NH polar (March) regions. The dashed black lines show the 1980 reference value for each latitude band. The shaded regions show 1-σ variability w.r.t. the MMM1S lines of the same colour. The top row shows the unadjusted modelled values and the bottom row shows the time series adjusted w.r.t. mean 1980-1984 observations and after application of a 10-point boxcar smoothing. Also shown are the merged SBUV observations.

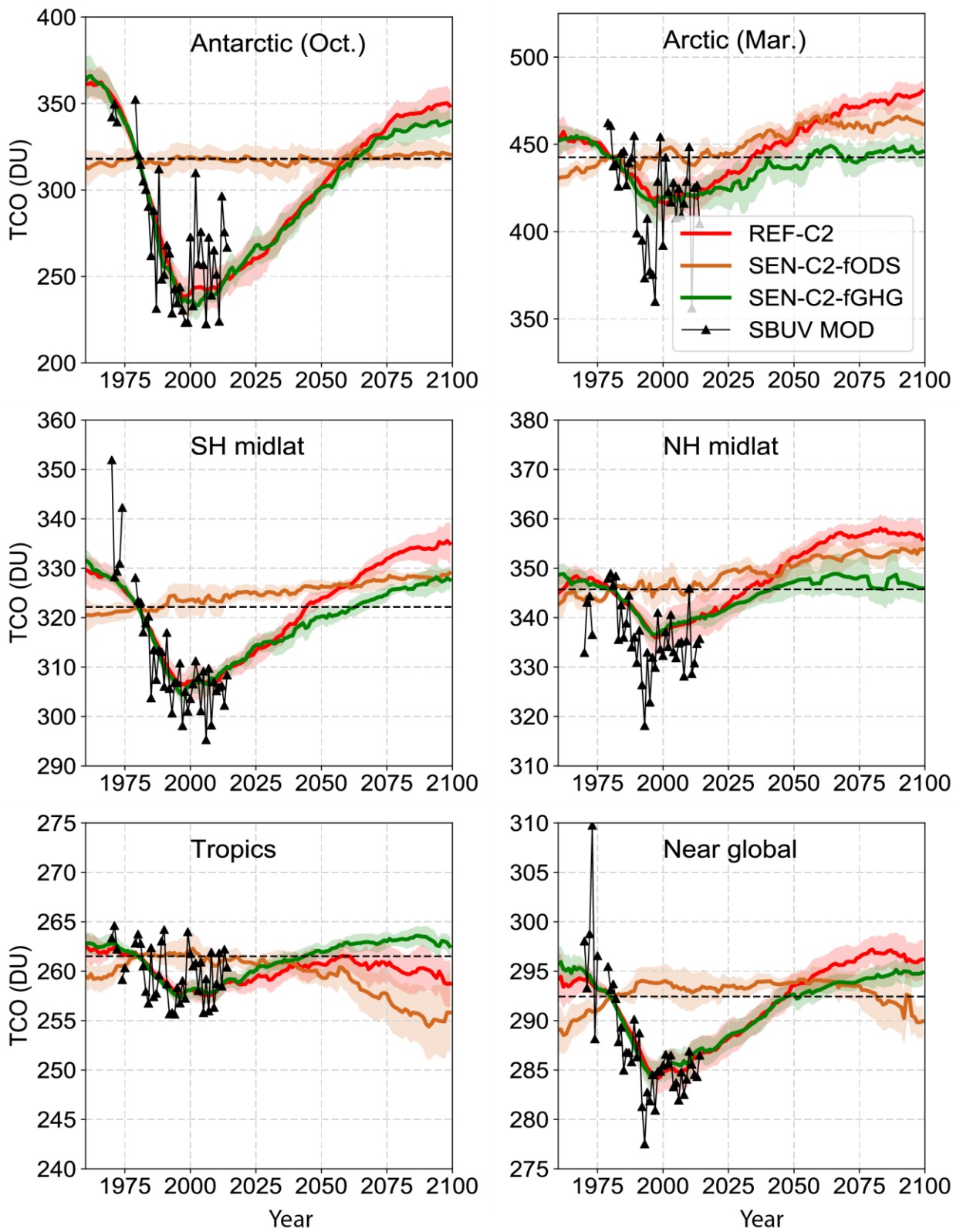

**Figure 3**. MMM1S total column ozone time series (DU) from REF-C2 (red), SEN-C2-fGHG (dark green), and SEN-C2-fODS (brown) simulations for five latitudinal bands and the near-global (60ºS-60ºN) mean (see main text). The dashed black lines show the 1980 reference value for each latitude band. Also shown are the merged SBUV observations.

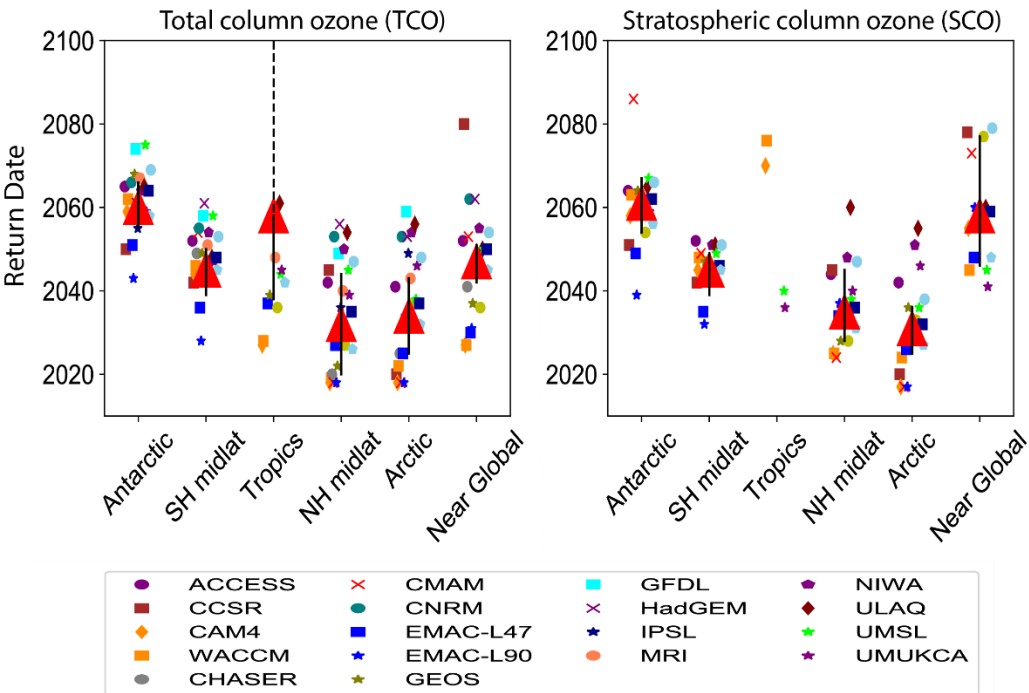

**Figure 4**. Estimated MMM1S return dates (red triangles) from the REF-C2 simulations for (left) total column ozone (TCO) and (right) stratospheric column ozone (SCO) for different latitude bands. The estimated 1-σ uncertainties are shown with vertical black lines. Estimates for individual models are shown with coloured dots. Some individual models do not predict a return of column ozone in the tropics and so this uncertainty is indicated by a dashed line.

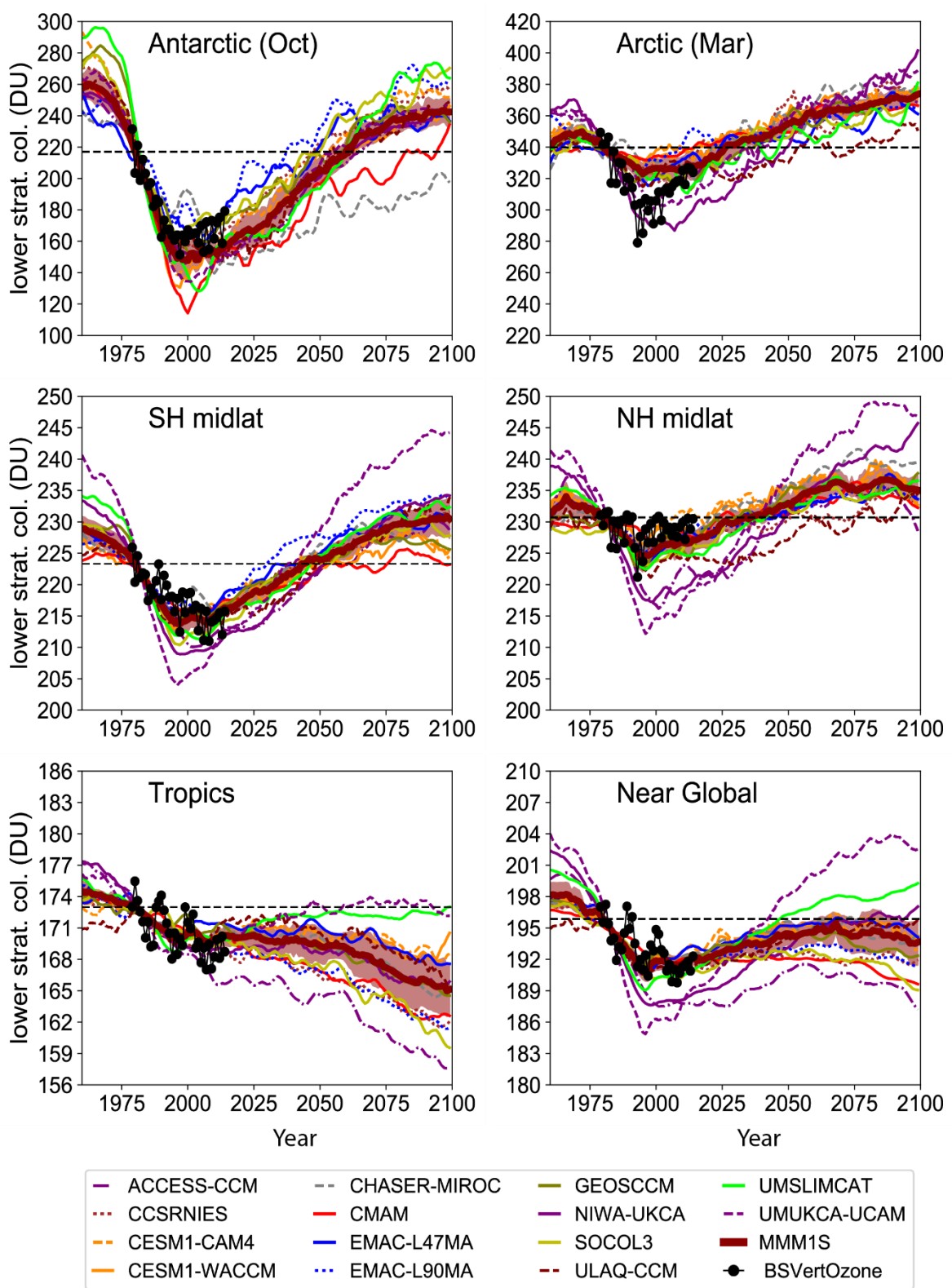

**Figure 5**. Evolution of partial column ozone (DU) for the lower stratosphere (tropopause - 10 hPa) from the REF-C2 simulations from 14 individual models, along with the MMM1S. Also shown are estimates of the partial column from the Bodeker Scientific Vertical Ozone (BSVertOzone) database, which is based on a compilation of satellite, balloon and ground-based measurements (*Bodeker et al.*, 2013).

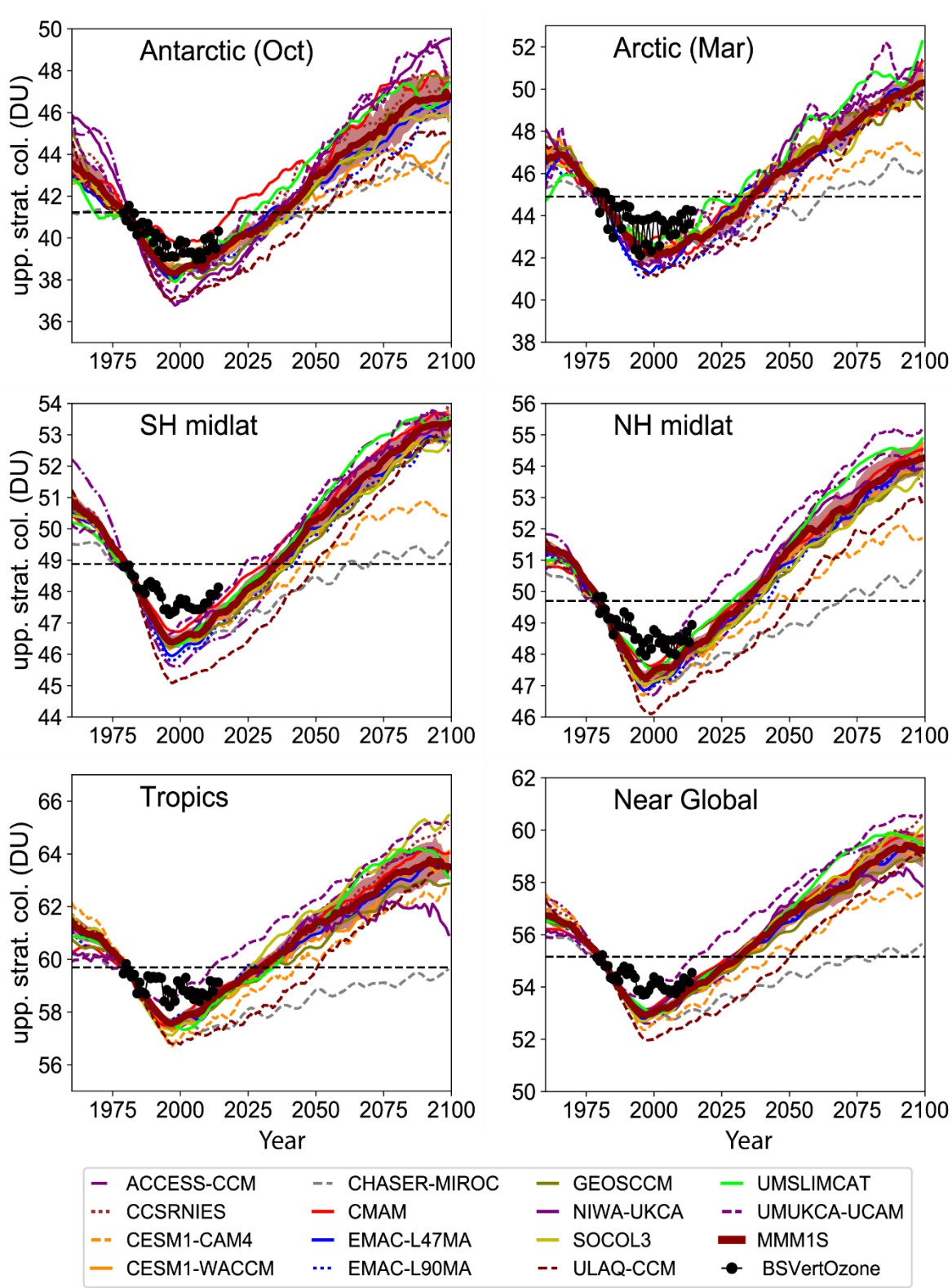

**Figure 6**. As **Figure 5** but for the upper stratosphere (≥ 10 hPa).

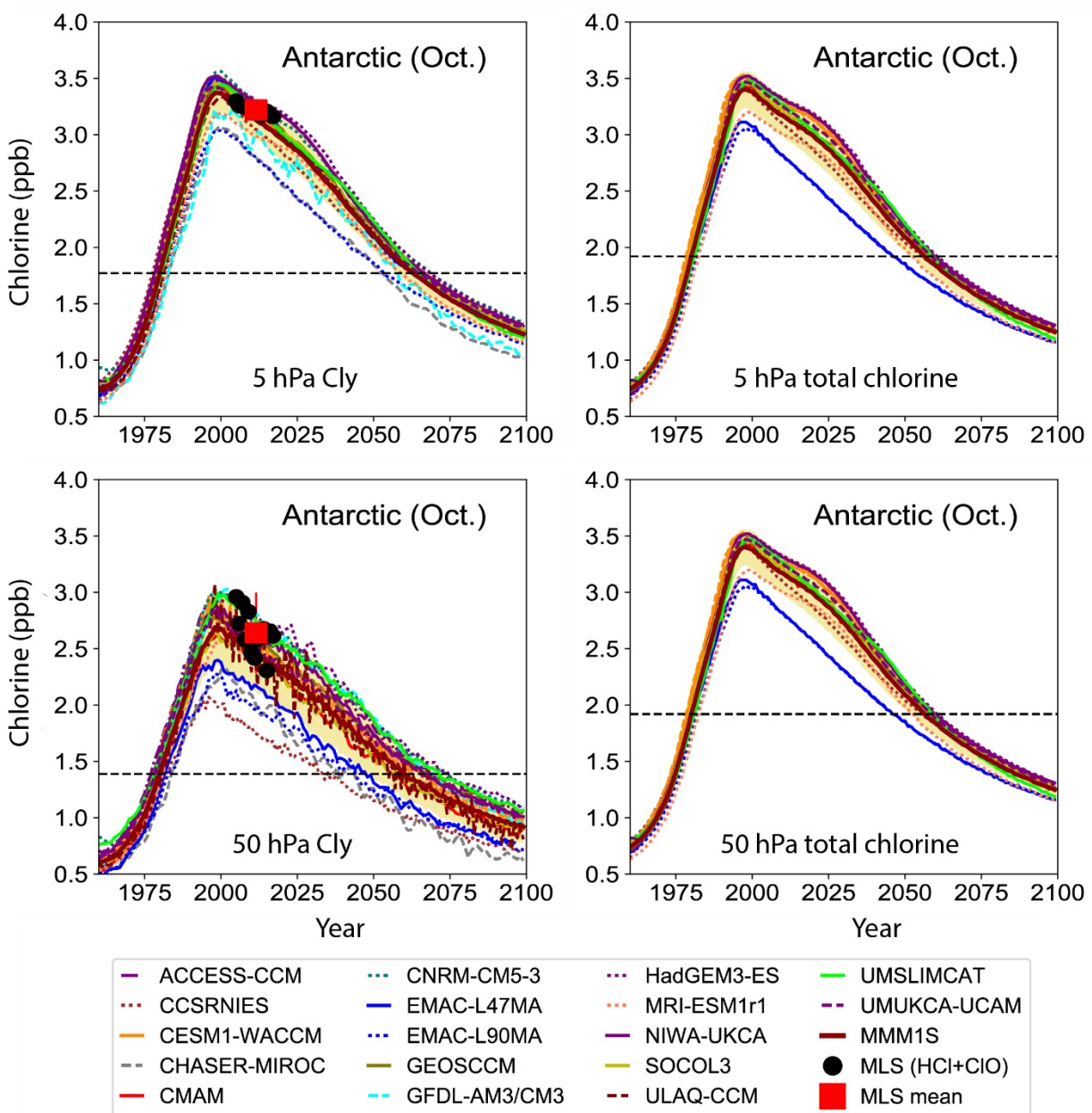

**Figure 7**. Evolution of (left) inorganic chlorine (Cly, ppb) and (right) total (organic + inorganic) chlorine from the REF-C2 simulations for 17 individual models and the MMM1S over the Antarctic in October at (top) 5 hPa and (bottom) 50 hPa. The dashed black lines show the 1980 reference value. Also shown in the left panels are observed October mean values of the sum of HCl and ClO from version 4 of the Microwave Limb Sounder (MLS) data (*Waters et al.*, 2006; *Livesey et al.*, 2017) from 2005 to 2017 (black dots) and the mean value over that period (red square). Note that not all models are plotted in the right-hand panels.

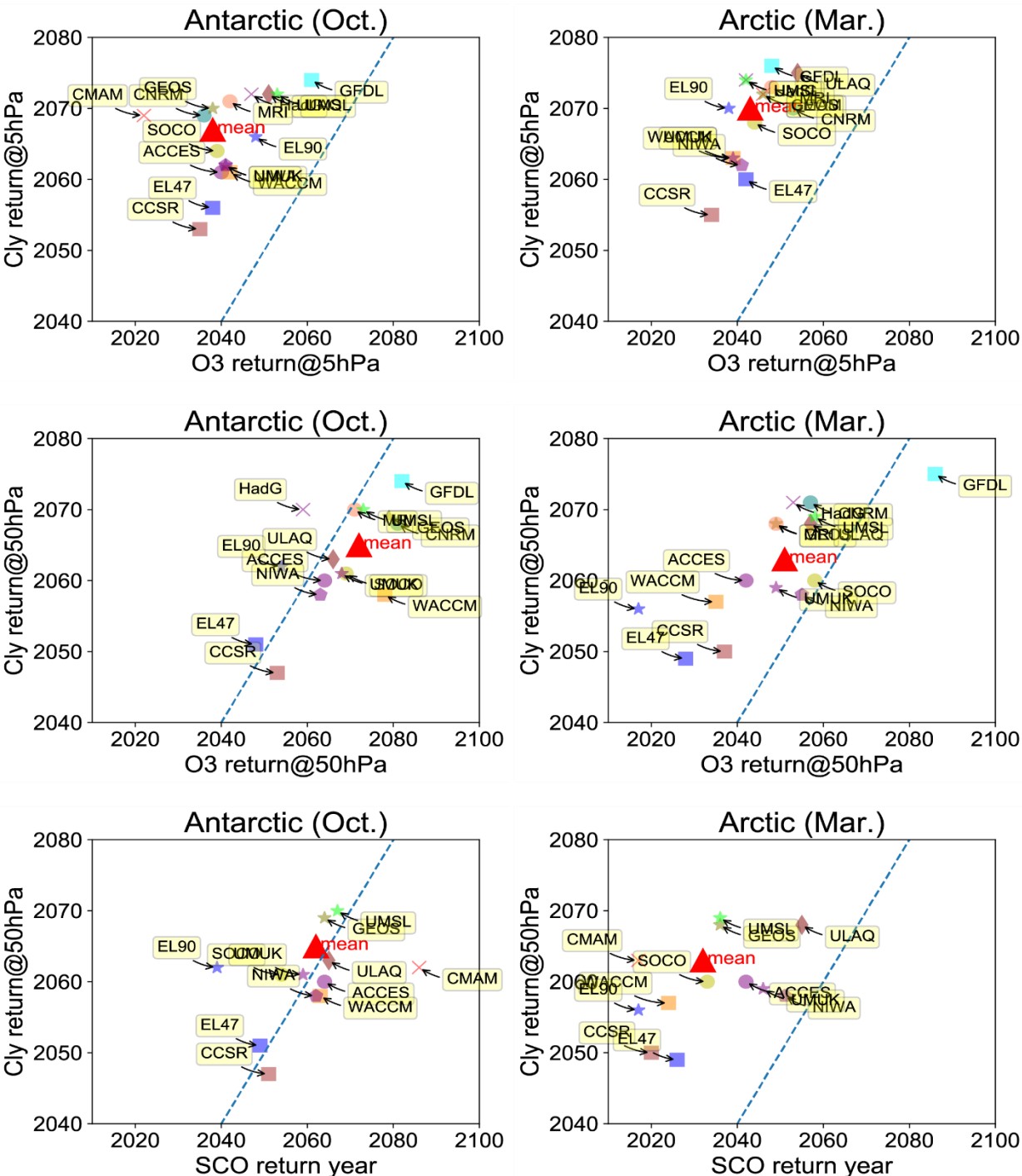

**Figure 8**. Correlation plots of ozone return dates against Cly return dates for (left) the Antarctic and (right) the Arctic from REF-C2 simulations for individual models and the MMM1S at (top) 5 hPa, (middle) 50 hPa and (bottom) stratospheric column (SCO). The red triangle is the multi-model mean. The dashed blue line is the 1:1 line between Cly and ozone return dates. The model symbols are the same as those used in **Figure 4**.

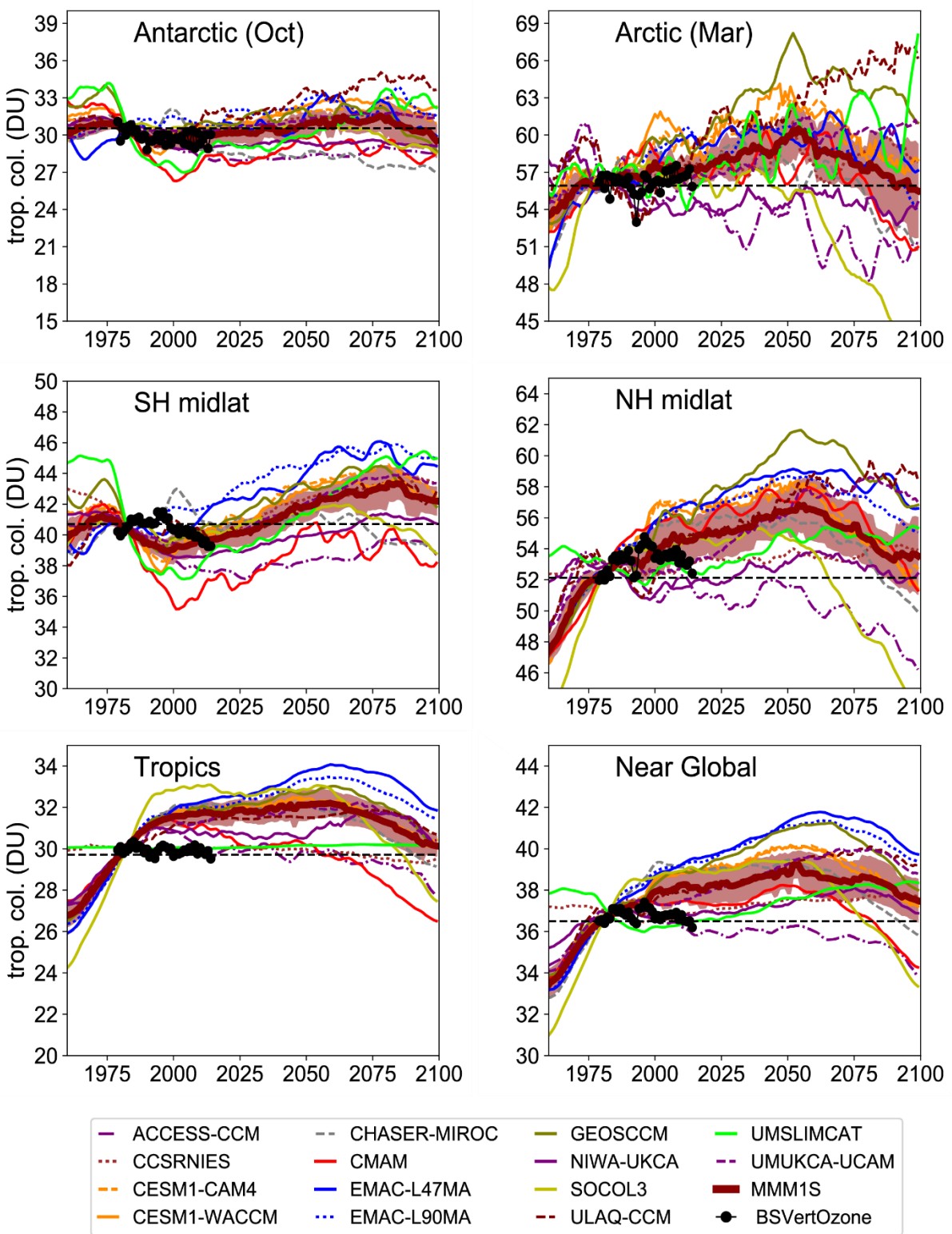

**Figure 9**. Evolution of tropospheric partial column ozone (DU) (surface - tropopause) from 14 individual models and the MMM1S for the REF-C2 simulations. Also shown is the tropospheric partial column ozone derived from BSVertOzone data.

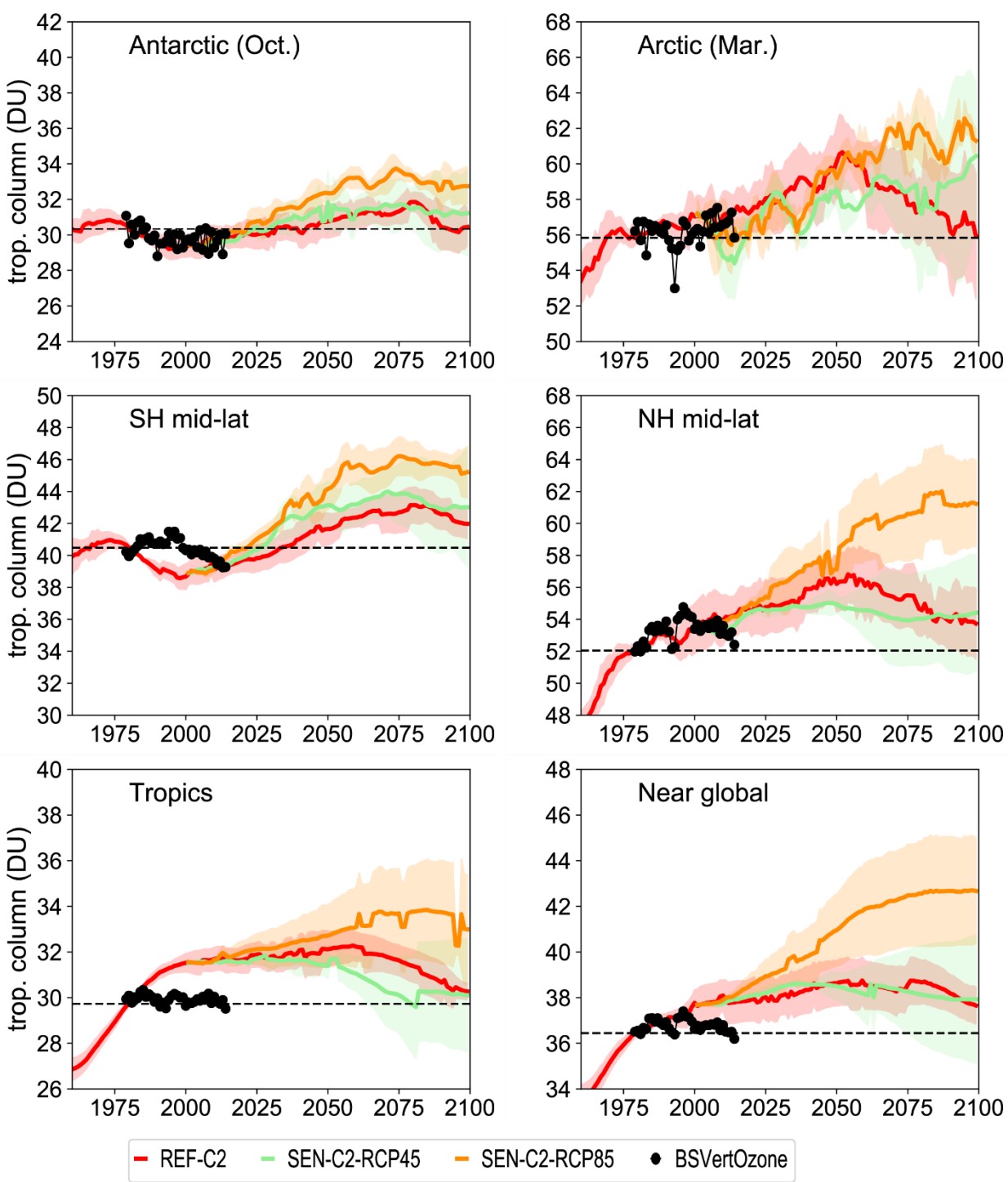

**Figure 10**. Evolution of tropospheric partial column ozone (DU) (surface-tropopause) MMM1S for REF-C2 and the RCP scenarios SEN-C2-RCP45 and SEN-C2-RCP85. Also shown is the tropospheric partial column ozone derived from BSVertOzone data.

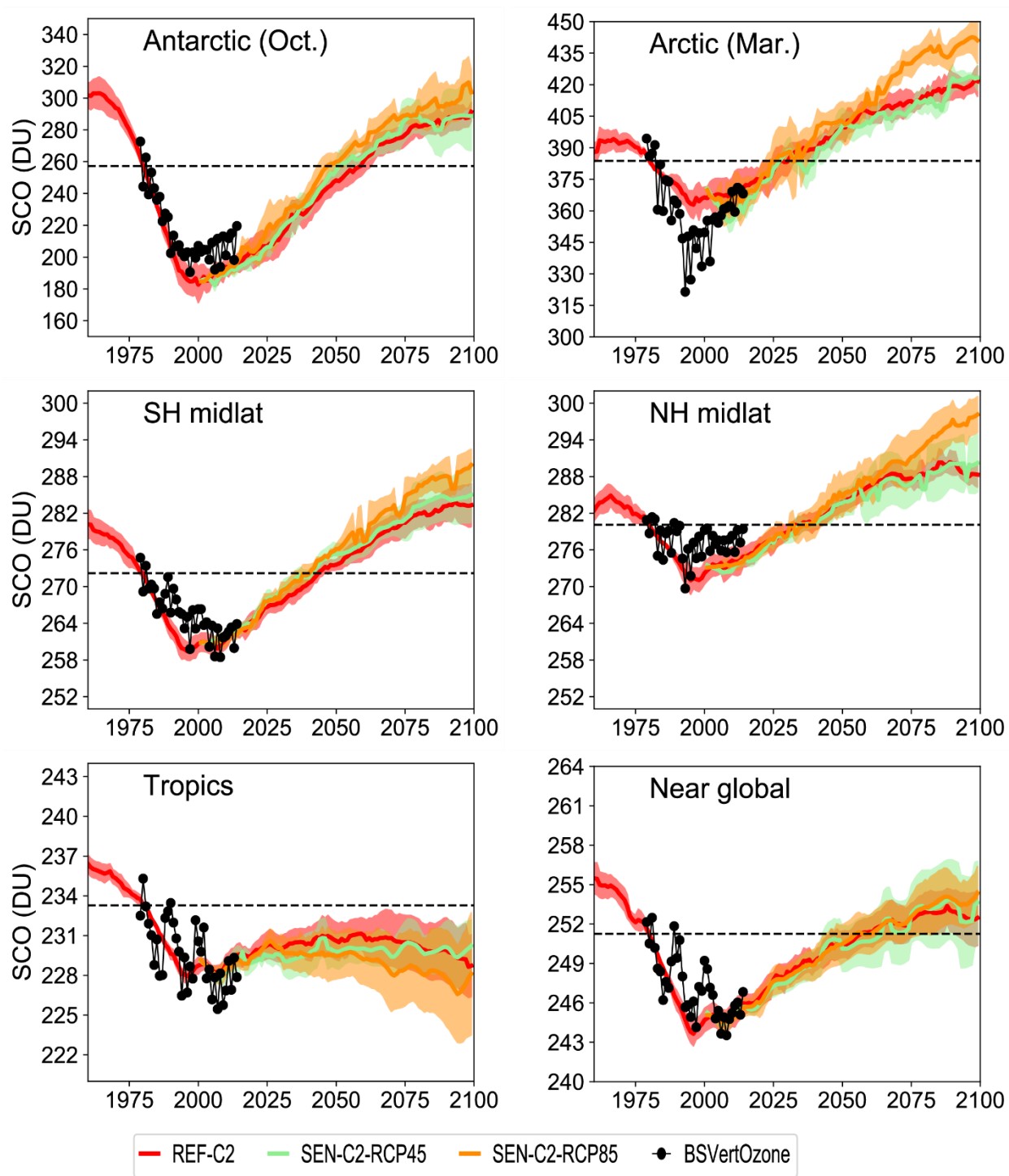

**Figure 11**. As **Figure 10** but for stratospheric column ozone (SCO).

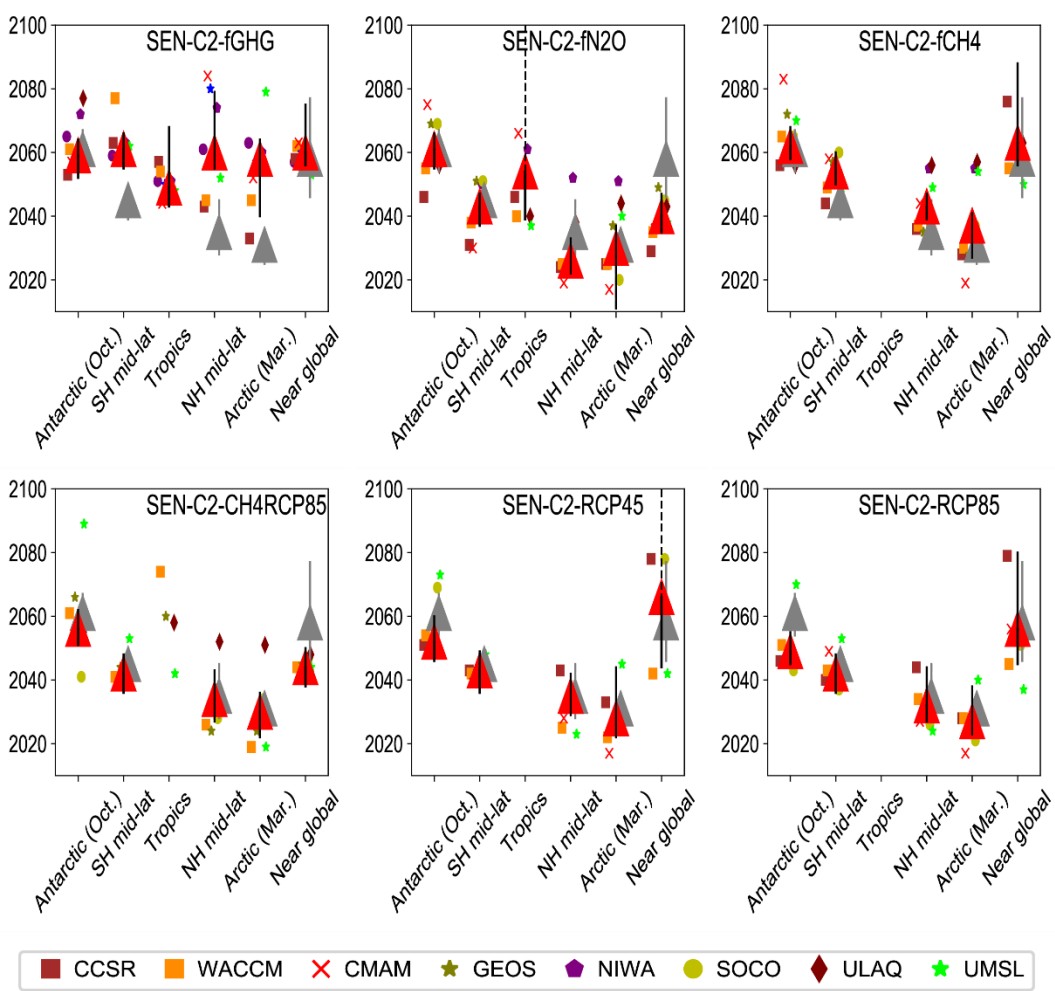

**Figure 12**. Estimated MMM1S return dates (red triangles) from stratospheric column ozone time series for (top) SEN-C2-fGHG, SEN-C2-fN2O and SEN-C2-fCH4 and (bottom) SEN-C2-CH4RCP85, SEN-C2-RCP45 and SEN-C2-RCP85. Estimated uncertainties are shown with vertical black lines. Grey triangles indicate SCO return dates from REF-C2 (**Figure 4**). Estimates for individual models are shown with coloured dots. Points with return dates (and uncertainties) that are greater than year 2100 are not shown.

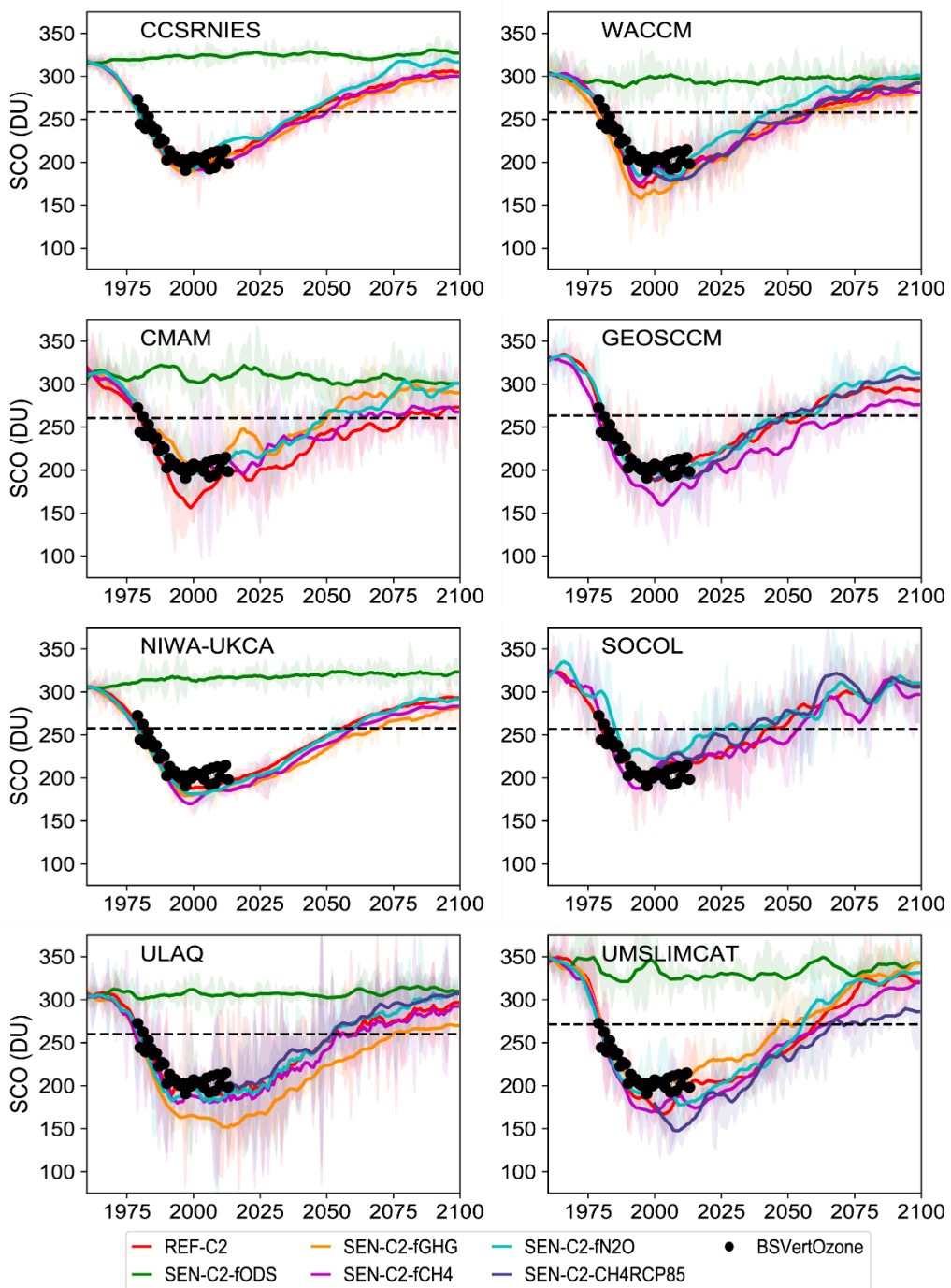

**Figure 13**. Evolution of Antarctic October mean stratospheric column ozone (DU) from 8 selected models for the REF-C2, SEN-C2-fCH4, SEN-C2-fN2O, SEN-C2-fODS, SEN-C2-fGHG and SEN-C2-CH4RCP85 simulations. Each panel gives the name of the model shown. The solid lines are the 10-year smoothed SCO for a given simulation. The shading on the lines shows either the standard deviation from an ensemble of realisations from that model, or the deviation from a 3-box smoothed line if only 1 realisation is available. Note that not all models have performed all simulations. Also shown in each panel is the SCO derived from BSVertOzone data (filled black circles).

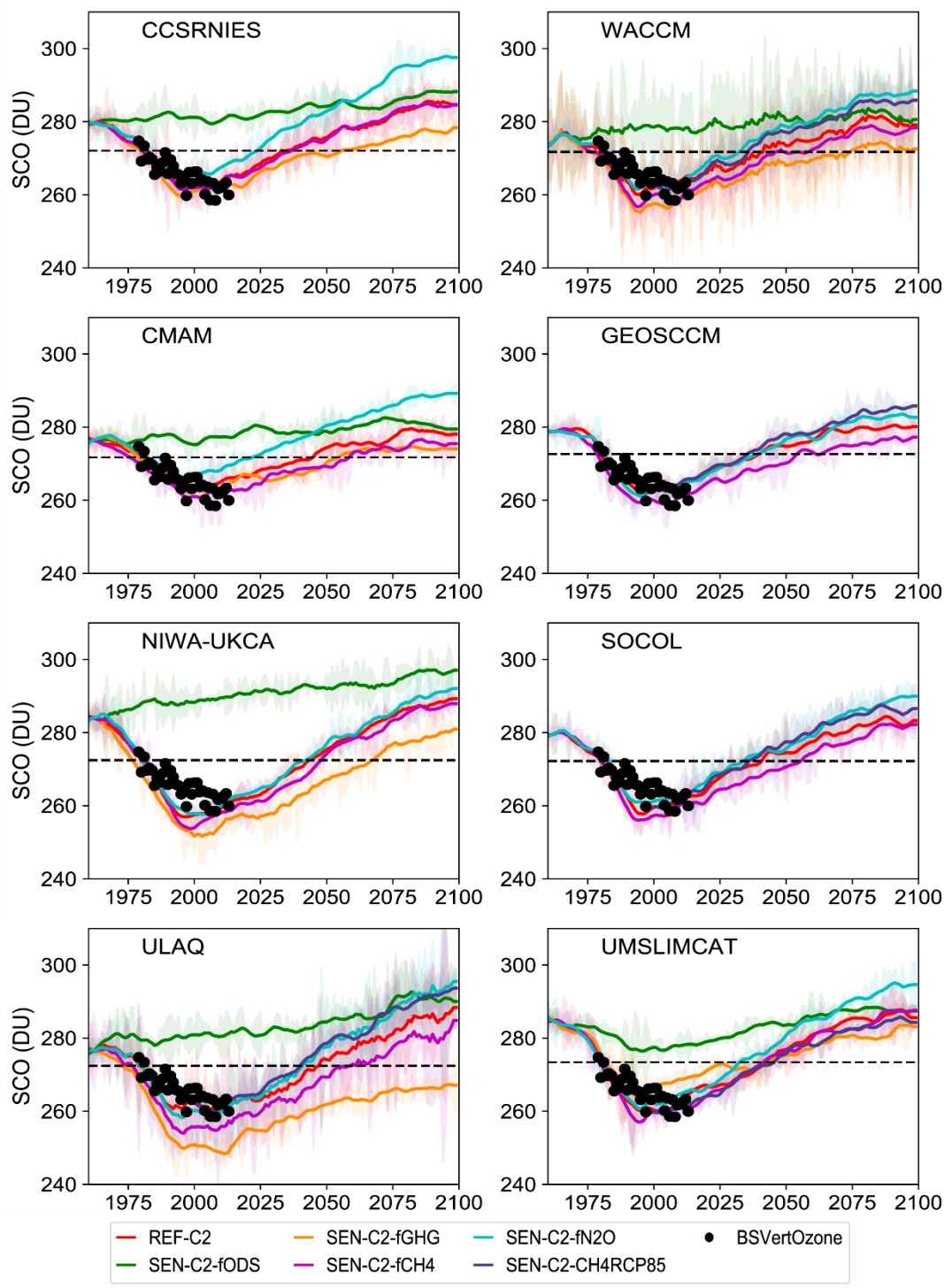

**Figure 14**. As **Figure 13** but for SH mid-latitude annual mean.