# Peer review of "Estimates of Ozone Return Dates from Chemistry-Climate Model Initiative Simulations"

_Atmospheric Chemistry and Physics, 2018_

## Referee Comment (RC1) · Anonymous Referee #1 · 7 Apr 2018

This work presents a detailed analysis of the ozone return dates from Chemistry-Climate Model Initiative (CCMI) simulations. The authors concluded that there exist strong regional differences in the future trend of total column ozone and its return date. The paper is well written and the results obtained in this study are useful for the research community. I would recommend publication with minor revision. The specific comments are listed bellow

Page 6, L15: Is a 10-point boxcar smoothing necessary? And whether this smoothing has an impact on the estimates of the return dates?

Page 7, L27: I cannot see the shading in Fig.1. If have, the shaded region is hardly detectable.

[Figure]

Page 7, L33: 'of the adjusted models' is misleading. It should be adjusted time series.

Page 8, L2: This conclusion may be right for zonal mean TCO, but sea-ice loss may affect zonally asymmetric TCO trends (Zhang et al., 2018). The authors should give some comments about why there is significant difference between REC-C1 and REF-C1SD during the 1990s in the Arctic for the adjusted time series (Fig.2d), which is not seen in the unadjusted series (Fig.2b). Zhang, J., Stratospheric ozone loss over the Eurasian continent induced by the polar vortex shift, 2018, Nature Communications, 9(1):206

Page 8, Line 5: The Antarctic and global ozone recovery rate before 2047 in REF-C2 is nearly the same as that in SEN-C-fGHG (Fig.3). But this feature is not seen for the other four latitudes. It is understandable that GHG has a little impact on Antarctic ozone; however, it is strange that GHG doesn't affect global mean ozone significantly before 2050. Does the tropical ozone loss cancel the extratropical ozone recovery?

P8, L20-22: Since the decline of the tropical ozone column is mainly due to transport, there should be a corresponding increase in the mid-high latitude TCO. Is it justified that the decline of the global TCO after about 2080 is mostly due to the decline of the tropical TCO?

Page 8, L35-40: It is interesting that the return dates in this study are all later than those detected from CCMval-1,2 and CMIP5 simulations. Do authors have any comments on this result? Is it resulted from the methodology used in this work?

Page 9 L10: The authors should provide more information of the vertical pressure of BSVertOzone. Did you interpolate the observed ozone profile onto the vertical pressure of CCMs and integrate the modeled and observed partial ozone column using the data at the same pressure levels?

Page 9, L13-14: This sentence is hard to understand. Please rephrase.

Page 9ïïijŇL36-38ïïijŽ The sentence is fragile. Please rephrase.

Page 9, L46: is»>its

Page 10ïïjŇ L12ïïjŽnotably to climate »>notably by climate change.

Page 11, L38-39: Wang et al (2017) pointed out that the effects of N2O increases on the stratospheric ozone are altitude dependent and GHG dependent. Wang W. et al (2014): Stratospheric ozone depletion from future nitrous oxide increase. ACP. 14, 12967-12982, 2014,. doi:10.5194/acp-14-12967-2014

Page 12, L20-L39: I suggested that the author could move the discussion regarding GHG to the Section 4.4.

Page 12, L45: Do you mean the red line and black dots for the three models in Fig.13? I don't think they are accurate, but the GEOSCCM and SOCOL simulations are better.

Page 13 L28,32: SOC->SCO

Page 13, L36-39: The authors argued (P8, L20-22) that a decrease in tropical ozone column contributes a decline in the global TCO after 2080. Here, the authors suggested that the dynamical transport process has no significant impact on the return date of global TCO. Which argument is correct?

Page 14 L25: but->by

---

## Referee Comment (RC2) · Anonymous Referee #2 · 13 Apr 2018

This paper details ozone return dates from the CCMI model intercomparison. As such, the new estimates of the return dates are the primary new finding of the paper and most of the other results discussed are previously known results. The paper is clearly written and outlining the latest results on ozone return dates is an important task for the upcoming ozone assessment. I have a number of comments but most are relatively minor and thus I recommend publishing the paper after the comments are adequately addressed.

As stated above most of the results outside of the specific details that are derived from the models simulations were already known. It would be good if you could highlight the new results from this paper in the abstract and conclusion and state more clearly the results that support earlier work.

[Figure]

It is important to state more prominently that since the models all use mixing ratio boundary conditions an important source of uncertainly has not been included. Thus, the uncertainty given are likely underestimated.

The resolution of the figures is poor, making them hard to read. This should be fixed in the final versions.

Page 2, lines 16-18: Are your later return dates within the uncertainly estimates given in the 2014 Ozone Assessment? This would be interesting to add.

Page 2, line 38: Change "also changes ozone return" to "also increases the ozone return" so the reader knows the direction of the change.

Page 3, line 5: I think you mean tropospheric chlorine and bromine peaked in 1993 and 1997, although I'll leave it to you to double check.

Page 3, line 16: The statement "therefore requires" is unjustified from what comes before. I'm not saying it is does not "require" 3D models, but just that you have not presented evidence that would support the claim.

Page 3, lines 47 to end of paragraph: The "faster removal of ODSs" is largely un-captured by the models here since they have fixed mixing ratio BCs. I say "largely uncaptured" since while the ODS loss term in the stratosphere is affected, it does not affect the surface mixing ratio as it should. This needs to be made clearer since this discussion is likely to mislead the uninformed reader.

Related to this effect it would be useful to have a figure of the CFC-11 lifetime (or some other long lived tracer) as a function of time from all the models.

Page 7, line 17: Using the period 1980-1984 is a bit unfortunate since there is signif-icant ozone loss during this period, especially in the Antarctic region. Depending on how you have done the calculation this will bias your return dates to earlier values. Since you have model data starting much earlier I would suggest using 1978-1982 in-stead, or at least discussing the sensitivity of your results are to this choice. I suspect

you made this choice due the availability of SBUV data but it should be possible to derive an adjustment for the data for your figures.

Page 7, line 28: I don't see any shading. Is this in reference to figure 1?

Page 9, lines 10-35: It is curious that nearly all of the model simulations are below the data. Any idea why? Seems worthy of mention and speculation.

Page 10, lines 1-9: The variation shown on figure 7 at 5 hPa is worrisome, a fact that should be highlighted in the paper. At 5 hPa most of the organic chlorine should be liberated (as can be seen by the similarity of the left and right panels) and thus both the plots should be close to the surface values with a 2-4 year lag to account for the age of air. The peak should be close to the peak in total chlorine in the surface concentrations and the values during the falloff should be very close to the surface values (since a 2-4 year shift is a small change). Thus, the models that are outliers on this plot are evidently not conserving chlorine and their results throughout the paper should be in question. This needs to be stated.

Page 10, line 40: Title should probably be "Sensitivity of ozone return to GHG concentrations and climate change"

Page 10, line 47: The tropospheric impact of CH4 has been pointed out in many papers before Morgenstern et al. 2018, so why chose this reference.

Page 11, line 4-6: Actually, the effect of GHG is as comparable in the Antarctic to the other regions. It is just harder to see because the scale of the chlorine depletion is so much larger. From you graph, I estimate a 20, 30, 5, and 8 DU change between RCP45 and RCP85 scenarios at 2100 in the 1st four panels. Thus, the Antarctic appears to be second largest instead of "small". This makes sense since the effect of the GHG is primarily above the ozone hole and thus should be similar. The main complication to this is the increased importance of Cl+CH4 in the ozone hole.

Page 11, lines 14-17: You state what is on figure 12 but say nothing of what it tells the

reader. Either discuss or remove.

Page 12, line 3: Change "chemically inert" to "chemically inert in the troposphere and stratosphere" since $CO_2$ is broken down in the mesosphere and above.

Page 12, line 10-11: Change "most important" to "most important for dynamical changes" or something similar.

Page 12, line 48-49: As above I disagree with this statement. It seems comparable to me looking at your plots if one adjusts to the greatly different scales. If you plotted the difference between the scenarios it would be clear.

Page 13, lines 39-44: The fact that the global value for the return date is seemingly inconsistent with the different latitude regions implies that it is poorly constrained and you can conclude little to nothing about the effect of $N_2O$ changes.

Page 14, line 39: Again, it is not the weakest but only less evident.

Page 15, line 7-9: You need to point out there are still serious issues with the chemical and/or transport schemes in some models.

Page 15, line 19-41: The points made in these final two paragraphs are important and should be made in the abstract as well.

Page 28: You should mention the shaded regions in the caption.

Page 30: Add the abbreviations SCO and TCO to the caption in the correct places to help the reader who doesn't read the paper.

————————————————

---

## Referee Comment (RC3) · Anonymous Referee #3 · 16 Apr 2018

The manuscript uses the latest CCMI simulations to derive new estimates of the ozone layer return dates. The study is an important update of the existing CCMVal2 evaluations and will provide valuable input for the next WMO ozone assessment. The paper is clearly written and well-structured and I recommend publication after the following comments have been addressed.

1) The authors adjust the model results to avoid biases when comparing to historical data and to reduce the spread in the predictions of the ozone column. However, this method can introduce new errors if the bias is not constant over time but process-related and time dependent. The manuscript misses a discussion of possible shortcomings of this method. What do the return dates look like before the adjustment? Is the mean return date the same and only the spread is reduced or are the models

on average over/underestimating the atmospheric ozone abundance? In this context it is not clear what the difference between Figure 1a and 1b is. From the text and the captions it sounds like, the only difference is the adjustment to the 1980-1984 values, but the lines look like the models have been smoothed as well. Furthermore, it is also not clear what the impact of excluding models outside the 1 sigma uncertainty range is. Is this only reducing the uncertainty or also changing the mean values?

2) The comparison of the modeled lower and upper stratospheric ozone columns with the BSVertOzone data set gives large differences for some regions (even after the bias adjustment). I miss a discussion of possible reasons for the over- and underestimation of ozone loss and possible implications for the projected return dates.

3) In parallel with the CCMI activities, the stratospheric ozone community has undergone large efforts to provide updates of the ozone profile trends from observational data sets (e.g. Steinbrecht et al., 2017). How do the models compare to these new results? Do they agree on the upper stratosphere ozone recovery quantified for the 2000-2016 time period?

4) How different are the chlorine comparisons if HCl+ClO instead of Cly is used in order to have a consistent comparison between models and measurements? How does the amount of stratospheric bromine differ from model to model and how do such differences impact the return dates? Why not use the EESC instead of Cly?

Page 8, line 41: Do you mean 2046?

---

## Author Comment (AC1) · 5 May 2018

We thank the reviewer#3 for his/her useful comments. These are repeated below in italics, followed by our responses after the '»'.

The manuscript uses the latest CCMI simulations to derive new estimates of the ozone layer return dates. The study is an important update of the existing CCMVal2 evaluations and will provide valuable input for the next WMO ozone assessment. The paper is clearly written and well-structured and I recommend publication after the following comments have been addressed.

1) The authors adjust the model results to avoid biases when comparing to historical data and to reduce the spread in the predictions of the ozone column. However,

this method can introduce new errors if the bias is not constant over time but process related and time dependent. The manuscript misses a discussion of possible short-comings of this method. What do the return dates look like before the adjustment? Is the mean return date the same and only the spread is reduced or are the models on average over/underestimating the atmospheric ozone abundance? In this context it is not clear what the difference between Figure 1a and 1b is. From the text and the captions it sounds like, the only difference is the adjustment to the 1980-1984 values, but the lines look like the models have been smoothed as well. Furthermore, it is also not clear what the impact of excluding models outside the 1 sigma uncertainty range is. Is this only reducing the uncertainty or also changing the mean values?

» Figure 1 shows the impact of the adjustment on the return dates (compare panels a and b) and this is discussed in Section 4.1. The aim of the adjustment is to improve the visualisation of the different models on the same plots will not impact the return date for any single model; it is a case of just displacing the ozone curve on the y axis so the return to the 1980 value will still occur at the same time. The impact of excluding the models outside the 1-sigma uncertainty is shown by comparison of the MMM and MMM1S results (e.g. Figure 1 and Table 3). Overall there is a small impact of the return date but a decrease in the uncertainty. Yes, the reviewer is correct that there also additional smoothing between Fig 1a and Fig 1b (and Fig 2a and 2b). This is the application of the10-point boxcar smoothing. Text had been added to clarify this.

2) The comparison of the modeled lower and upper stratospheric ozone columns with the BSVertOzone data set gives large differences for some regions (even after the bias adjustment). I miss a discussion of possible reasons for the over- and underestimation of ozone loss and possible implications for the projected return dates.

» We have updated the BSVertOzone dataset,which was revised before the submission of Hassler et al. (2018). The lower stratospheric comparisons are improved. We have added some sentences to point out the remaining upper stratospheric difference and to say that we do not think differences in this region will greatly affect column ozone

return dates.

3) In parallel with the CCMI activities, the stratospheric ozone community has undergone large efforts to provide updates of the ozone profile trends from observational data sets (e.g. Steinbrecht et al., 2017). How do the models compare to these new results? Do they agree on the upper stratosphere ozone recovery quantified for the 2000-2016 time period?

» Results from CCMI models have been included in the LOTUS report. That report is not finalised (and so we cannot cite it yet) but there will be information in there. A separate analysis in this current paper is beyond the scope of our current work.

4) How different are the chlorine comparisons if HCl+ClO instead of Cly is used in order to have a consistent comparison between models and measurements? How does the amount of stratospheric bromine differ from model to model and how do such differences impact the return dates? Why not use the EESC instead of Cly?

» The MLS ClO + HCl is a good surrogate for total Cly for the regions that we compare. The differences between the observed sum and true Cly will be small compared to the model-model spread that we wish to illustrate. Also, we do not have the CCMs sampled as the MLS local time for a true comparison of ClO. We now present the modelled Bry loadings in the Supplementary Material. As more models provided Cly output compared to Bry it is simpler to use Cly than EESC, and avoids the complication of selecting an alpha value for the models.

Page 8, line 41: Do you mean 2046?

» Yes, corrected.

---

## Author Comment (AC2) · 5 May 2018

We thank the reviewer#2 very much for his/her insightful comments. These are repeated below in italics, followed by our responses after the '»'.

This paper details ozone return dates from the CCMI model intercomparison. As such, the new estimates of the return dates are the primary new finding of the paper and most of the other results discussed are previously known results. The paper is clearly written and outlining the latest results on ozone return dates is an important task for the upcoming ozone assessment. I have a number of comments but most are relatively minor and thus I recommend publishing the paper after the comments are adequately addressed.

[Figure]

As stated above most of the results outside of the specific details that are derived from the models simulations were already known. It would be good if you could highlight the new results from this paper in the abstract and conclusion and state more clearly the results that support earlier work.

» We appreciate the comment but this is not a general point that was raised by the other reviewers. While dealing with the other review comments we have rewritten the abstract and conclusions to add content and clarify some points. However, based on the other review comments, we think that the overall main results of the paper are clear. The paper does present up-to-date, state-of-the-art estimates of return dates in support of the WMO/UNEP Ozone Assessment and also (especially with the revisions) discusses the uncertainties in the concept and numerical value of return dates.

It is important to state more prominently that since the models all use mixing ratio boundary conditions an important source of uncertainly has not been included. Thus, the uncertainty given are likely underestimated.

» This is stated in the Introduction, which now includes the following sentence: "This shortening of the ozone recovery was also found by Morgenstern et al. (2018) for the models represented in this study, although it is important to note that the use of surface mixing ratios in studies largely removes the feedback between circulation changes and ODS return dates."

The resolution of the figures is poor, making them hard to read. This should be fixed in the final versions.

» Yes, it will be. The figures appeared fine in our submitted pdf but the conversion to the ACP pdf (adding the journal information on each page) seemed to degrade the figures.

Page 2, lines 16-18: Are your later return dates within the uncertainly estimates given in the 2014 Ozone Assessment? This would be interesting to add.

[Figure]

» The uncertainty range from our and previous studies (e.g. WMO (2014)) are given in Table 3. It is difficult to compare uncertainty ranges concisely in the abstract so we have added some text indicating that the previous (WMO (2014)) best estimates are often outside the uncertainty ranges that we quote.

Page 2, line 38: Change "also changes ozone return" to "also increases the ozone return" so the reader knows the direction of the change.

» OK. We have changed the word to 'lengthens'.

Page 3, line 5: I think you mean tropospheric chlorine and bromine peaked in 1993 and 1997, although I'll leave it to you to double check.

» Yes, corrected, thank you.

Page 3, line 16: The statement "therefore requires" is unjustified from what comes before. I'm not saying it is does not "require" 3D models, but just that you have not presented evidence that would support the claim.

» OK. We have added some words about the possible changes to polar vortex dynamics to the end of the previous paragraph to better motivate the need for 3-D models (as opposed to just 2-D).

Page 3, lines 47 to end of paragraph: The "faster removal of ODSs" is largely uncaptured by the models here since they have fixed mixing ratio BCs. I say "largely uncaptured" since while the ODS loss term in the stratosphere is affected, it does not affect the surface mixing ratio as it should. This needs to be made clearer since this discussion is likely to mislead the uninformed reader.

» OK, we have added some text to note this.

Related to this effect it would be useful to have a figure of the CFC-11 lifetime (or some other long lived tracer) as a function of time from all the models.

» Unfortunately the information needed for this comparison was not saved by the CCMI

models. However, as the reviewer points out, the use of vmr boundary conditions constrains the models to more similar chlorine distributions more than would be the case for differeing lifetimes. The SPARC lifetime assessment did compile this information from a subset of models used here. The comparison of Cly shown in this paper does depend on the modelled halocarbon lifetimes.

Page 7, line 17: Using the period 1980-1984 is a bit unfortunate since there is significant ozone loss during this period, especially in the Antarctic region. Depending on how you have done the calculation this will bias your return dates to earlier values. Since you have model data starting much earlier I would suggest using 1978-1982 instead, or at least discussing the sensitivity of your results are to this choice. I suspect you made this choice due the availability of SBUV data but it should be possible to derive an adjustment for the data for your figures.

» We don't think that this will bias the return dates. The mean value for the period 1980-84 is used to estimate the adjustment, but the adjusted model values still vary during this period. Also, 1980 is the specific year chosen for the reference date. The fact that ozone loss may occur more strongly later in this period should not matter. In any case, satellite observations are not available for 1978.

Page 7, line 28: I don't see any shading. Is this in reference to figure 1?

» Yes. The shading is very pale and was affected by the conversion to the ACPD pdf. This will be made clearer in the final figure versions.

Page 9, lines 10-35: It is curious that nearly all of the model simulations are below the data. Any idea why? Seems worthy of mention and speculation.

» We have updated the BSVertOzone dataset,which was revised before the submission of Hassler et al. (2018). The lower stratospheric comparisons are improved. We have added some sentences to point out the remaining upper stratospheric difference and to say that we do not think differences in this region will greatly affect column ozone

return dates.

Page 10, lines 1-9: The variation shown on figure 7 at 5 hPa is worrisome, a fact that should be highlighted in the paper. At 5 hPa most of the organic chlorine should be liberated (as can be seen by the similarity of the left and right panels) and thus both the plots should be close to the surface values with a 2-4 year lag to account for the age of air. The peak should be close to the peak in total chlorine in the surface concentrations and the values during the falloff should be very close to the surface values (since a 2-4 year shift is a small change). Thus, the models that are outliers on this plot are evidently not conserving chlorine and their results throughout the paper should be in question. This needs to be stated.

» The models do show a variation in Cly at 5 hPa, but it is not possible from this plot alone to conclude that they do not conserve mass. The differences could be due to incorrect scenarios of the long-lived halocarbons or differences in the treatment of short-lived sources. Nevertheless, we agree that we should not be seeing models with chlorine a lot different to the prescribed scenarios. We have added the sentences: "Moreover, in the upper……of around 2-4 years".

Page 10, line 40: Title should probably be "Sensitivity of ozone return to GHG concentrations and climate change"

» Section 4.5 deals with the impact of the GHGs CH4 and N2O. Section 4.4 is about the effect of climate in general and therefore to distinguish it from 4.5 we would like to keep the title as it is.

Page 10, line 47: The tropospheric impact of CH4 has been pointed out in many papers before Morgenstern et al. 2018, so why chose this reference.

» We have added additional references to the earlier papers of Shindell et al (2009) and Eyring et al (2013b).

Page 11, line 4-6: Actually, the effect of GHG is as comparable in the Antarctic to the

other regions. It is just harder to see because the scale of the chlorine depletion is so much larger. From you graph, I estimate a 20, 30, 5, and 8 DU change between RCP45 and RCP85 scenarios at 2100 in the 1st four panels. Thus, the Antarctic appears to be second largest instead of "small". This makes sense since the effect of the GHG is primarily above the ozone hole and thus should be similar. The main complication to this is the increased importance of Cl+CH4 in the ozone hole.

» We agree that the absolute variation is similar (or larger) in the Antarctic (Oct) relative to the other regions/periods but we wish to compare the relative variations. We have modified text to include: "The relative change is smaller in the Antarctic, where recovery is largely determined by Cly loading, but larger in all other regions. However, the absolute changes between, for example, the Antarctic (October) and Arctic (March) are similar".

Page 11, lines 14-17: You state what is on figure 12 but say nothing of what it tells the reader. Either discuss or remove.

» We understand the reviewer's point for the first mention of Figure 12. Figure 12 (along with Table 4) is meant to be a summary figure of the discussion surrounding "Figure 11". We have added additional information on how this figure relates to the MMM1S return date for SEN-C2-RCP45 and SEN-C2-RCP85 relative to REF-C2 (Figure 4). This figure is also discussed several times in sections 4.4 (Sensitivity of ozone return to climate change) and 4.5 (Sensitivity of ozone return to methane and nitrous oxide)

Page 12, line 3: Change "chemically inert" to "chemically inert in the troposphere and stratosphere" since CO2 is broken down in the mesosphere and above.

» OK, we have added 'below about 60km'.

Page 12, line 10-11: Change "most important" to "most important for dynamical changes" or something similar.

» OK, we have added 'for dynamical processes'.

Page 12, line 48-49: As above I disagree with this statement. It seems comparable to me looking at your plots if one adjusts to the greatly different scales. If you plotted the difference between the scenarios it would be clear.

» The statement above was for Figure 13, not Figure 11. Here eight models were shown with identical y-axis ranges. To make the sentence clearer, we have modified the sentence to read: "In summary, when one examines the relative impact on the ozone return date across the eight models from the four SEN-C2 scenarios, there is not a consistent pattern. Therefore, the result suggests that the Antarctic region is not sensitive to the perturbations presented in this work."

Page 13, lines 39-44: The fact that the global value for the return date is seemingly inconsistent with the different latitude regions implies that it is poorly constrained and you can conclude little to nothing about the effect of N2O changes.

» The effect of N2O on ozone is complicated, but we can still draw conclusions. We have changed the text in question to read:

In this comparison, one would expect that SEN-C2-fN2O with 1960 abundances of N2O would bring forward the SCO recovery date. This is certainly true for the near-global (annual) average comparison, where the MMM1S SEN-C2-fN2O SCO recovery date is shortened by ∼20 years relative to the REF-C2 case. This is mostly due to a shortening of the return date in the tropics; at mid-high latitudes there is little change. As mentioned above the future rise in N2O can lead to significant increases in lower-stratospheric ozone, particularly for regions where the loss rate of ozone due to halogens exceeds that due to NOx prior to the perturbation of N2O. The effect of N2O on ozone varies as a function of latitude and altitude (Wang et al., 2014), complicating the sensitivity to the ozone return date to variations in N2O (Morgenstern et al., 2018).

The new text now includes a citation to the Wang et al. (2014) paper, pointed out by reviewer #1, which had not originally been cited.

Page 14, line 39: Again, it is not the weakest but only less evident.

» OK, we have changed to 'least evident'.

Page 15, line 7-9: You need to point out there are still serious issues with the chemical and/or transport schemes in some models.

» OK, text has been added.

Page 15, line 19-41: The points made in these final two paragraphs are important and should be made in the abstract as well.

»OK. Some of the points were already in the abstract (e.g. impact of N2O and CH4, also starting with text 'As noted by previous studies...', and comments on how to use models in future assessments). The points that were not mentioned relate to uncertainties in scenarios and models. We have added sentences on this to the abstract.

Page 28: You should mention the shaded regions in the caption.

» OK, text has been added.

Page 30: Add the abbreviations SCO and TCO to the caption in the correct places to help the reader who doesn't read the paper.

» OK, done.
* * *

---

## Author Comment (AC3) · 5 May 2018

We thank the reviewers very much for their comments. These are repeated below in italics, followed by our responses after the '»'.

Anonymous Referee #1

This work presents a detailed analysis of the ozone return dates from Chemistry-Climate Model Initiative (CCMI) simulations. The authors concluded that there exist strong regional differences in the future trend of total column ozone and its return date. The paper is well written and the results obtained in this study are useful for the research community. I would recommend publication with minor revision. The specific comments are listed below.

[Figure]

Page 6, L15: Is a 10-point boxcar smoothing necessary? And whether this smoothing has an impact on the estimates of the return dates?

» We perform the smoothing to remove interannual variability – our aim is to determine a robust estimate of the return date which reflects the average behaviour of the atmosphere rather than on short-term dynamical variability. This point is now made in new text in the Introduction on what we are aiming for with a return date.

Page 7, L27: I cannot see the shading in Fig.1. If have, the shaded region is hardly detectable.

» The shading is there in the original figure but is very pale. It is within about 25 DU of the MMM line. Unfortunately it almost completely disappears in the ACPD pdf conversion. We will aim to improve this in the final figures.

Page 7, L33: 'of the adjusted models' is misleading. It should be adjusted time series.

» OK. We have changed this to 'forecasts provided by the adjusted models'.

Page 8, L2: This conclusion may be right for zonal mean TCO, but sea-ice loss may affect zonally asymmetric TCO trends (Zhang et al., 2018). The authors should give some comments about why there is significant difference between REC-C1 and REFC1SD during the 1990s in the Arctic for the adjusted time series (Fig.2d), which is not seen in the unadjusted series (Fig.2b). Zhang, J., Stratospheric ozone loss over the Eurasian continent induced by the polar vortex shift, 2018, Nature Communications, 9(1):206

» OK. We have added a reference to the Zhang et al. paper. The Arctic REF-C1 v REF-C1SD difference in the 1990s is likely related to the series of cold Arctic winters in this period which resulted in large column ozone loss. The mean of the free-running REF-C1 simulations do not capture this. We have added some text on this.

Page 8, Line 5: The Antarctic and global ozone recovery rate before 2047 in REF-C2 is nearly the same as that in SEN-C2-fGHG (Fig.3). But this feature is not seen for

the other four latitudes. It is understandable that GHG has a little impact on Antarctic ozone; however, it is strange that GHG doesn't affect global mean ozone significantly before 2050. Does the tropical ozone loss cancel the extratropical ozone recovery?

» Please note that the polar regions are not included in the 'near global' mean plotted in Figure 3. In terms of the other regions, yes there is some cancellation of mid-latitudes (fGHG later) and the tropics (fGHG earlier). Please also note that the stratospheric column values are given in Table 4 and this shows similar behaviour, indicating that it is a stratospheric effect.

Page 8, L20-22: Since the decline of the tropical ozone column is mainly due to transport, there should be a corresponding increase in the mid-high latitude TCO. Is it justified that the decline of the global TCO after about 2080 is mostly due to the decline of the tropical TCO?

» The reviewer raises an important point (see also his/her comment below on SCO). Again, please note that the 'global' referred to near-global, i.e. excluding the polar regions where ozone continues to increase through 2100. Figure 9 shows the partial tropospheric ozone column. This does also show a decrease after 2080, by around 2-3 DU at low-mid latitudes, which is similar to the TCO decline. Moreover, the tropospheric decrease is larger in the NH than the SH, similar to the TCO. Therefore, we have clarified the text on this point (and on the SCO below).

Page 8, L35-40: It is interesting that the return dates in this study are all later than those detected from CCMval-1,2 and CMIP5 simulations. Do authors have any comments on this result? Is it resulted from the methodology used in this work?

» We have added a more detailed discussion on this. Factors include the update WMO halocarbon scenario and different climate forcings.

Page 9 L10: The authors should provide more information of the vertical pressure of BSVertOzone. Did you interpolate the observed ozone profile onto the vertical pressure

of CCMs and integrate the modeled and observed partial ozone column using the data at the same pressure levels?

» We have added detail to the paper: "BSVertOzone spans 70 pressure levels that are approximately 1km apart (878.4hPa to 0.046hPa). For the calculation of the partial columns, ozone was interpolated to the exact boundaries of the partial columns from the two closest BSVertOzone pressure levels, and then ozone was integrated between the determined levels. The boundaries for the partial columns were defined as follows: tropospheric column (surface-tropopause), lower stratospheric column (tropopause - 10hPa), upper stratospheric column (10hPa and above; for BSVertOzone this means up to 0.046hPa). The tropopause pressure was defined as 100hPa in the tropics (20S-20N), 150hPa in the mid-latitudes (20-60N/S), and 200hPa in the polar regions (60-90N/S). CCM partial columns were integrated between the same partial column boundaries, but directly from the CCM pressure levels. No additional interpolation of CCM ozone profiles or BSVertOzone profiles was performed."

Page 9, L13-14: This sentence is hard to understand. Please rephrase.

» We have rewritten this.

Page 9 L36-38 The sentence is fragile. Please rephrase.

» We have rephrased this.

Page 9, L46: isÂż>its

» Corrected.

Page 10 L12 notably to climate Âż>notably by climate change.

» We have modified the text, and split into two sentences.

Page 11, L38-39: Wang et al (2014) pointed out that the effects of N2O increases on the stratospheric ozone are altitude dependent and GHG dependent. Wang W. et al (2014): Stratospheric ozone depletion from future nitrous oxide increase. ACP. 14,

12967-12982, 2014, doi:10.5194/acp-14-12967-2014.

» OK. We have added a new sentence with citation to Wang et al and Revell et al, after the citation to an early Crutzen paper.

Page 12, L20-L39: I suggested that the author could move the discussion regarding GHG to the Section 4.4.

» We understand the suggestion, but have decided to leave the text in place, because Section 4.4 focuses on the effect of climate change on ozone for a single GHG scenario, whereas Section 4.5 explores the sensitivity of future O3 to how CH4 and N2O evolve.

Page 12, L45: Do you mean the red line and black dots for the three models in Fig.13? I don't think they are accurate, but the GEOSCCM and SOCOL simulations are better.

» Apologies. The text was not clear or accurate (and was left over from a previous version of the figure). In fact, many models perform well. The text has been updated.

Page 13 L28,32: SOC->SCO

» Corrected.

Page 13, L36-39: The authors argued (P8, L20-22) that a decrease in tropical ozone column contributes a decline in the global TCO after 2080. Here, the authors suggested that the dynamical transport process has no significant impact on the return date of global TCO. Which argument is correct?

» P8 referred to total column ozone (TCO). This page discusses the stratosphere only (SCO), the difference being the tropospheric column (see Figure 9). We have clarified the earlier TCO discussion to mention the tropospheric contribution.

Page 14 L25: but->by

» Corrected.